# ACTIVE LEARNING FOR DECISION TREES WITH PROVABLE GUARANTEES

**Arshia Soltani Moakhar**
Univeristy of Maryland
asoltan3@umd.edu

**Tanapoom Laoaron**
Univeristy of Maryland
tlaoaron@umd.edu

**Faraz Ghahremani**
Univeristy of Maryland
farazgh@umd.edu

**Kiarash Banihashem**
Univeristy of Maryland
kiarash@umd.edu

**MohammadTaghi Hajiaghayi**
Univeristy of Maryland
hajiagha@umd.edu

## ABSTRACT

This paper advances the theoretical understanding of active learning label complexity for decision trees as binary classifiers. We make two main contributions. First, we provide the first analysis of the **disagreement coefficient** for decision trees—a key parameter governing active learning label complexity. Our analysis holds under two natural assumptions required for achieving polylogarithmic label complexity: (i) each root-to-leaf path queries distinct feature dimensions, and (ii) the input data has a regular, grid-like structure. We show these assumptions are essential, as relaxing them leads to polynomial label complexity. Second, we present the first general active learning algorithm for binary classification that achieves a **multiplicative error guarantee**, producing a $(1 + \epsilon)$-approximate classifier. By combining these results, we design an active learning algorithm for decision trees that uses only a **polylogarithmic number of label queries** in the dataset size, under the stated assumptions. Finally, we establish a label complexity lower bound, showing our algorithm's dependence on the error tolerance $\epsilon$ is close to optimal.

## 1 INTRODUCTION

Active learning is a machine learning paradigm that seeks to minimize the labeling effort required to train a model by strategically selecting the most informative data points for labeling Ren et al. (2022). Unlike traditional passive learning, which relies on randomly labeled data, active learning operates on an unlabeled dataset and iteratively selects a sample to *query* its label which tailors the selection process to focus on examples that contribute the most to improve model performance. Labeling complexity becomes a significant challenge in scenarios where annotation requires human expertise—particularly in domains like medical applications, where labeling cannot be outsourced to crowdsourcing platforms but instead relies on skilled professionals. Given the limited availability and high cost of such experts, active learning emerges as an invaluable solution in domains where acquiring labeled data is both expensive and time-consuming. Examples include medical diagnosis Budd et al. (2021), autonomous driving Feng et al. (2019), webpage classification Hanneke (2014), and natural language processing Schröder & Niekler (2020); Zhang et al. (2022). By reducing the labeling cost, active learning has become a cornerstone of efficient model development in data-intensive fields Settles (2012).

Decision trees are extensively utilized in machine learning because they inherently perform feature selection Xu et al. (2014); Banihashem et al. (2023), offer interpretability Gilpin et al. (2018), and achieve strong practical performance with minimal computational expense. These properties have made decision trees a core component in ensemble methods such as random forests Breiman (2001) and XGBoost Chen & Guestrin (2016), which are among the most popular algorithms in supervised learning tasks. While active learning has been applied to decision trees in various practical contexts Ma et al. (2016); Wang et al. (2010), the existing research in this area often lacks a rigorous theoretical foundation. This gap highlights the need for a deeper understanding of the theoretical aspects of applying active learning principles to decision tree learning.

This paper addresses a significant gap in the theoretical foundations of active learning by providing the first rigorous analysis of its sample complexity for decision trees. Our analysis centers on the *disagreement coefficient*, a key parameter in active learning theory that, until now, had not been analyzed for the decision tree class. The importance of bounding this coefficient lies in its direct impact on label efficiency, as it appears in many active learning algorithms. For example Hanneke (2014) get the following label complexity:

$$\theta(\frac{\nu^2}{\epsilon_{\text{additive}}^2} + \log \frac{1}{\epsilon_{\text{additive}}})(d \log \theta + \text{Log}\left(\frac{\log(1/\epsilon_{\text{additive}})}{\delta}\right))$$

This investigation reveals two critical assumptions required to derive a polylogarithmic bound on this coefficient: that each node in the decision tree must test a feature dimension distinct from its ancestors, and that the input data exhibits structural regularity (which we model as a grid). We prove that without these assumptions, the disagreement coefficient is not effectively bounded, leading to polynomial sample complexity. Our analysis culminates in the following bound:

**Theorem 1.1.** *Consider a decision tree classification task over a dataset $S$ of $n$ points. Let the input space be $X = \{(a_1, \ldots, a_{dim}) \mid \forall i, a_i \in \mathbb{N}, a_i \leq w\}$ for some $w$. If every node in a tree tests a feature dimension distinct from its ancestors and the tree height is at most $d$, then the disagreement coefficient is upper bounded by $\theta = O(\ln^d(n))$ and lower bounded by $\theta = \Omega(c(\ln(n) - c')^{d-1})$ where $c = \frac{2^{-dim}}{d^{d-1}d!}$ and $c' = \ln(4)dim$.*

Next, we propose the first active learning algorithm for binary classification tasks on discrete datasets that achieves a multiplicative error bound, Algorithm 2. In our framework, an algorithm is given $n$ unlabeled data points and can adaptively query their binary labels. The objective is to return a $(1 + \epsilon)$-approximate classifier with probability at least $1 - \delta$. A classifier is $(1 + \epsilon)$-approximate if its error is at most $1 + \epsilon$ times that of the optimal classifier in the class. Our primary focus is to minimize the algorithm's *label complexity*, which is the total number of queries it performs. The performance of Algorithm 2 is captured by the following theorem:

**Theorem 1.2.** *For any binary classification task, Algorithm 2 returns a $(1 + \epsilon)$-approximate classifier with probability greater than $1 - \delta$. It does so using*

$$O\left(\ln(n)\theta^2(V_H \ln \theta + \ln \frac{\ln n}{\delta}) + \frac{\theta^2}{\epsilon^2}(V_H \ln \frac{\theta}{\epsilon} + \ln \frac{1}{\delta})\right)$$

*queries, where $n$ is the dataset size, $V_H$ is the VC dimension of the classifier space, and $\theta$ is the disagreement coefficient.*

The adoption of the multiplicative error model in classification tasks is a key strength of this work. It enables stronger control over the classifier's accuracy compared to additive error models, providing greater flexibility to achieve desirable error rates based on what is achievable. For instance, in realizable settings, where the optimal classifier has zero error, this approach guarantees perfect classification—a capability beyond the reach of additive models, which $\epsilon$-off regardless of the optimal classifier error. While the multiplicative framework has been extensively explored in the context of active learning for regression (see, e.g., Musco et al. (2022); Derezinski et al. (2018); Parulekar et al. (2021); Chen & Price (2019); Chen & Derezinski (2021); Gajjar et al. (2023; 2024); Chen et al. (2022)), our work is the first to introduce an algorithm for multiplicative error in classification. This aligns with broader trends in computer science, such as approximation algorithms and competitive analysis, where multiplicative error models are standard.

A crucial motivation for our work is the inadequacy of existing additive error algorithms for the multiplicative setting. A natural approach might be to adapt an additive algorithm by setting its error parameter $\epsilon_{\text{additive}}$ relative to an estimate of the optimal error $\eta$ (i.e., $\epsilon_{\text{additive}} = \epsilon\eta$). However, this strategy is fundamentally flawed. Estimating $\eta$ with sufficient accuracy to provide a meaningful guarantee itself requires a number of label queries that is inversely proportional to $\eta$. Consequently, the label complexity would become dependent on an unknown and potentially very small quantity, making the required number of labels $\Omega(n)$. Alternative strategies, such as iteratively guessing and verifying $\eta$, face the same bottleneck at the verification step. In Appendix E, we formalize this argument, demonstrating that any such adaptation is inherently label-inefficient. This highlights the

need for a fundamentally different approach, like the one we propose, that is designed to be agnostic to the magnitude of the optimal error.

Our central result is a new label complexity bound for actively learning decision trees, which we derive by combining the two main contributions of this paper. By applying our general multiplicative-error Algorithm 2 to the decision tree class and using our novel bound on the disagreement coefficient, we achieve a query complexity that is polylogarithmic in the dataset size. This result holds under the previously discussed assumptions on tree structure and data distribution. The algorithm's performance depends on the maximum tree depth, denoted by $d$, and the feature dimensionality, dim, as formalized in our main theorem:

**Corollary 1.3.** *Let* $X = \{(x_1, x_2, \ldots, x_{dim}) \mid \forall i, x_i \leq w, x_i \in \mathbb{N}\}$ *be a set with a binary labeling. Algorithm 2 returns a* $(1 + \epsilon)$*-approximate classifier of an optimal decision tree that each node operates on a data dimension distinct from those used by its ancestors. The algorithm requires at most the following number of queries:*

$$
O\left( \ln^{2d+2}(n)\Big( 2^d(d + \ln dim)d + \ln\frac{1}{\delta} \Big) \ + \frac{\ln^{2d}(n)}{\epsilon^2}\Big( 2^d(d + dim)\ln\frac{\ln^d(n)}{\epsilon} + \ln\frac{1}{\delta} \Big) \right).
$$

To highlight the efficiency of our algorithm, we also establish lower bounds for the label complexity of any such active learning algorithm in Theorem 4.3 and showed that some terms, like $\epsilon$, can only experience logarithmic improvements.

To summarize, our contributions are as follows:

- The first active learning algorithm for multiplicative error budget in classification.
- The first label complexity bound for active decision tree learning.
- Proving the necessity of the uniform-like assumption and the constraint that each node on root-to-leaf paths operates on a unique dimension for achieving poly-logarithmic label complexity in active decision tree learning.
- The first label complexity lower bound for active stump learning on discrete datasets.

## 2 RELATED WORKS

**Realizable Active Learning for classification:** Numerous studies have examined active learning in the context of binary classification tasks. Some of these works assume the existence of a classifier with zero error El-Yaniv & Wiener (2010; 2012); Hanneke (2012); Hopkins et al. (2020b). In contrast, our approach does not rely on this assumption, which makes it more applicable to real-world scenarios where a perfect classifier is not guaranteed.

**Agnostic Active Learning for classification:** Some prior research has addressed active learning in agnostic settings, where no perfect classifier exists Balcan et al. (2006; 2007); Hanneke (2007); Dasgupta et al. (2007); Castro & Nowak (2008; 2006). Among these, algorithms based on disagreement-based active learning, such as $A^2$ Balcan et al. (2006), share similarities with our approach by maintaining a version space—a set of classifiers that initially includes the optimal classifier and is iteratively refined without excluding it. However, unlike prior work, our method is the first to exploit signals arising when the version space fails to shrink rapidly. We use this stagnation to lower-bound the error and leverage this lower bound to identify a $(1 + \epsilon)$-approximate classifier.

Notably, all these previous studies assume an additive error framework, guaranteeing that the classifier's error exceeds that of the optimal classifier by at most a fixed additive margin. In Appendix E, we examine the relationship between additive and multiplicative error frameworks and algorithms, demonstrating that existing additive approaches are unsuitable for multiplicative error settings and cannot be adapted to address our problem. For a comprehensive survey, see Hanneke (2014).

**Active Learning in Regression:** Active learning has been extensively studied in the context of linear regression and $\ell_p$ norm regression. Several papers aim to improve the label requirements of active learning and provide theoretical bounds on the minimum requirements Musco et al. (2022); Derezinski et al. (2018); Parulekar et al. (2021); Chen & Price (2019); Chen & Derezinski (2021);

Woodruff (2014); Sarlós (2006). Notably, Musco et al. (2022) investigates $\ell_p$ norm regression. Throughout our paper, we adopt the setup from Musco et al. (2022), and our algorithm returns a $(1 + \epsilon)$-approximate solution on a given discrete dataset where labels are arbitrary, without any assumptions on them.

**Theory of Decision Tree Learning:** The theoretical study of decision tree learning has been explored primarily from the perspective of time complexity in various specific contexts Ehrenfeucht & Haussler (1989); Mehta & Raghavan (2002); Blanc et al. (2019; 2022).While recent work has established sample complexity bounds for data-driven hyperparameter tuning of decision trees Balcan & Sharma (2024), there is no previous work that considers theoretical guarantees for the active learning label complexity of decision trees. Additionally, in the context of learning, existing works have analyzed different properties of decision trees, including their application to sparse feature recovery. Notably, works by Banihashem et al. (2023); Kazemitabar et al. (2017) focused on decision stump learning for regression problems, motivated by the challenge of sparse feature recovery. While these studies have contributed to our understanding of decision tree learning in settings such as sparse recovery and sample complexity, they largely overlook the domain of active learning within decision tree theory.

**Disagreement Coefficient of decision trees** The theoretical basis for active learning of decision trees was laid by Balcan et al. (2010), who showed that axis-parallel trees on continuous inputs in the $[0, 1]^n$ hypercube can be learned efficiently under the uniform distribution. Their proof relied on decomposing the class by leaf count and arguing that each subclass has a finite disagreement coefficient. However, they only asserted finiteness without giving a way to compute the coefficient or bound it quantitatively. Our work fills this gap by providing the first explicit calculation of the disagreement coefficient for decision trees on discrete domains, establishing the bound $\theta = O(\ln^d(n))$.

**Active Learning Using Stronger Queries:** To overcome the limitations of conventional active learning methods, several studies have considered active learning with stronger query models Hopkins et al. (2021); Kane et al. (2017); Hopkins et al. (2020a). For example, Hopkins et al. (2021) investigates the active learning of decision trees using queries that check whether two samples belong to the same leaf in the optimal decision tree. Similarly, Kane et al. (2017) shows that it is possible to learn a perfect half-space using only $\log(\text{dataset size})$ comparison queries, where the labeler answers which sample is more positive. However, these approaches assume realizable settings, where a perfect classifier exists. In contrast, our work focuses on a simpler and more practical query model that exclusively returns the label of a sample, without assuming realizability.

## 3 DISAGREEMENT COEFFICIENT IN DECISION TREES

The theoretical analysis of many active learning algorithms for binary classification relies on the *disagreement coefficient*, a parameter that measures the complexity of a hypothesis class Balcan et al. (2006); Hanneke (2014). Intuitively, it captures how many data points have uncertain labels within a set of plausible hypotheses. A smaller coefficient suggests that an active learning strategy can efficiently prune the version space. This section defines the disagreement coefficient and derives an upper bound for the class of decision trees.

### 3.1 FORMAL DEFINITIONS

Let $H$ denote the hypothesis class (e.g., decision trees of bounded depth) and $S$ a dataset of $n$ points.
**Definition 3.1** (Distance & Hypothesis Ball)**.** The *distance* between two hypotheses $h, h' \in H$ on $S$ is the fraction of points where they disagree:

$$D_S(h, h') := \tfrac{1}{n} \sum_{x \in S} \mathbb{I}(h(x) \neq h'(x)).$$

The *ball* of radius $r$ around $h \in H$ is the set

$$B_H(h, r) := \{h' \in H \mid D_S(h, h') \leq r\}.$$

**Definition 3.2** (Disagreement Region)**.** For $V \subseteq H$, the *disagreement region* is the set of points in $S$ where some pair of hypotheses in $V$ differ:

$$\text{DIS}_S(V) := \{x \in S \mid \exists h_1, h_2 \in V : h_1(x) \neq h_2(x)\}.$$

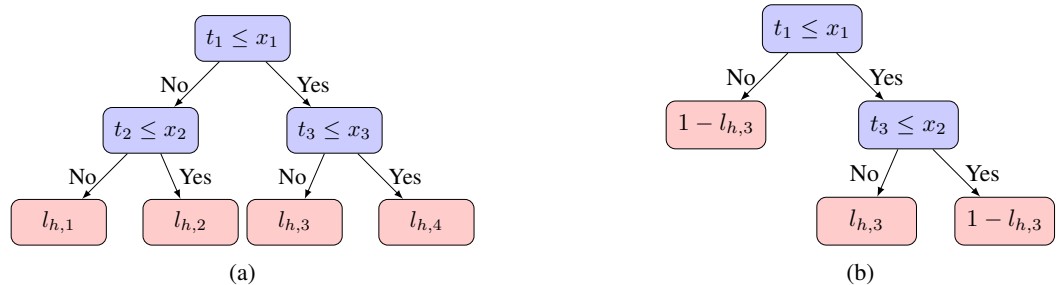

Figure 1: (a) A decision tree with $4$ leaves ($L = 4$). Leaf 1 uses dimensions $1, 2$ so $d_{h,1} = \{1, 2\}$. (b) LineTree$_{h,3}$ classifies all samples as $1 - L_{h,3}$ except those reaching leaf 3 of $h$.

These notions lead to the *disagreement coefficient*, which compares the size of the disagreement region to the radius of the corresponding hypothesis ball.

**Definition 3.3** (Disagreement Coefficient). For $h \in H$, the disagreement coefficient is

$$\theta_h := \sup_{r>0} \frac{|\mathrm{DIS}_S(B_H(h,r))|}{rn}.$$

The disagreement coefficient for $H$ is the worst-case value: $\theta := \sup_{h \in H} \theta_h$.

## 3.2 AN UPPER BOUND FOR DECISION TREES

Our first main result, Theorem 1.1, establishes an upper bound on the disagreement coefficient for decision trees under certain structural and distributional assumptions. The proof proceeds by decomposing a tree into simpler components, analyzing their disagreement properties, and then recombining the results, which leads us to define the notion of a LineTree.

Let $h$ be a decision tree. For each leaf $i$, let $l_{h,i}$ denote its label and $d_{h,i} \subseteq \{1, \ldots, \dim\}$ the set of dimensions tested along the path from the root to that leaf.

**Definition 3.4** (LineTree). For a tree $h$ and leaf $i$, the corresponding *line tree*, denoted LineTree$_{h,i}$, is a classifier that assigns label $l_{h,i}$ to all inputs reaching leaf $i$ in $h$, and the opposite label $1 - l_{h,i}$ otherwise. Figure 1 shows an example of a tree $h$ and its line tree LineTree$_{h,3}$.

To prove Theorem 1.1, we fix a tree $h$ and bound $\theta_h$ by analyzing $\frac{|\mathrm{DIS}(B_H(h,r))|}{nr}$ for all $r > 0$. Although $\mathrm{DIS}(B_H(h,r))$ seems to depend on all pairs of classifiers in $B_H(h,r)$, it can be expressed directly in terms of $h$. By Lemma D.1, $\mathrm{DIS}(B_H(h,r)) = \{x \mid \exists h' \in B_H(h,r) : h'(x) \neq h(x)\}$. We decompose this set according to the leaf $i$ that $x$ reaches in $h$ and the dimension set $d'$ of the leaf it reaches in $h'$. Hence, $\mathrm{DIS}(B_H(h,r))$ is the union over all $i$ and $d' \subseteq \{1, 2, \ldots, \dim\}$ of

$$\{x \mid \mathrm{LineTree}_{h,i}(x) = l_{h,i} \wedge \exists_{h' \in B_H(h,r),j} \, d_{h',j} = d' \wedge \mathrm{LineTree}_{h',j}(x) = l_{h',j} \wedge l_{h',j} \neq l_{h,i}\} \quad (1)$$

If we can replace $h' \in B_H(h,r)$ with an equation related to LineTree$_{h,i}$ and LineTree$_{h',j}$, then we can relate the analysis of trees to the analysis of line trees. In Lemma D.2, we achieve this by showing that if $l_{h,i} \neq l_{h',j}$, then $D_{S_i}(h, h')$ is larger than $D_{S_i}(\mathrm{LineTree}_{h,i}, \mathrm{LineTree}_{h',j})$ when $S_i$ is the set of data points that reaches leaf $i$ in $h$.

Then, we use Proposition 3.5 to show that the sets in Equation 1 each have a size of $O((\frac{2\ln(n)}{\dim})^d)$. Combining this with the fact that there are $L\binom{\dim}{d}$ sets in total, we can prove Theorem 1.1.

**Proposition 3.5.** *In a line tree classification task where each node decides based on one of the $d' \subseteq \{1, 2, \cdots, \dim\}$ input dimensions, different from all of its ancestors, and the tree height is less than d, with*

$$X = \{(a_1, \ldots, a_{dim}) \mid \forall_i \, a_i \in \mathbb{N}, a_i \leq w_i \leq w\}$$

*for some $w_i$ vector and $w$, the disagreement coefficient of a classifier which assigns the same labels to all data points is of $O\left((3\ln w)^d\right)$.*

Using the calculated disagreement coefficient and $V_H$ of decision tree in Lemma A.3 which is $2^d(d + \ln \dim)$, we can completes the proof of Corollary 1.3.

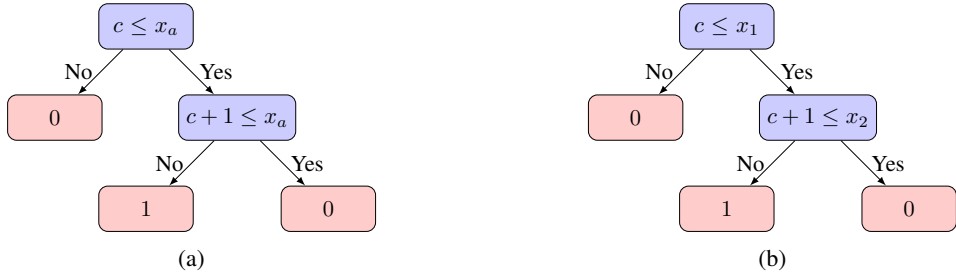

Figure 2: (a) A decision tree that assigns label 1, if and only if $x_a = c$. (b) A decision tree assigning label 1 only to $X_c$ when $X_i = \langle i, i, \cdots, i \rangle$.

### 3.3 NECESSITY OF ASSUMPTIONS

We now show that the assumptions in Theorem 1.1 are necessary. Without them, the disagreement coefficient becomes substantially larger.

**Theorem 3.6.** *If decision tree nodes are permitted to query the same dimension as their ancestors, the disagreement coefficient for trees of height $d \geq 2$ is $\theta = \Omega(n^{1/dim})$ for any dataset with $n$ distinct points.*

To prove this we first consider the constant classifier $h_0$ that labels all samples as 0. Since decision tree nodes can query the same dimension as their ancestors, it is possible to construct classifiers that are very close to $h_0$ but label an small portions of the data as 1. This allows us to build a set of classifiers within a small ball around $h_0$ whose disagreement region covers a large fraction of the dataset. An example of such classifier is presented in Figure 2a, which classifier only data points that have $x_a = c$ for some specific $a$ and $c$.

More specifically, we can construct such a set of classifiers within a radius of $r = 2n^{-1/\dim}$ around $h_0$. The resulting disagreement region, $\text{DIS}(B_H(h_0, r))$, can be shown to contain at least $\frac{2^{\dim}-1}{2^{\dim}} \cdot n$ data points. This large disagreement region within a small radius directly leads to a large disagreement coefficient. Calculating the ratio $\frac{|\text{DIS}|}{rn}$ with these values yields a lower bound of $\theta = \Omega(n^{1/\dim})$.

**Theorem 3.7.** *There exists a size $n$ dataset for which the disagreement coefficient of a binary decision tree classifier is $\Omega(n)$, even if nodes are restricted to unique dimensions per root-to-leaf paths.*

To prove this we first consider a dataset where all points lie on the line $x_1 = x_2 = \cdots = x_{\dim}$, e.g., $X_i = \langle i, i, \ldots, i \rangle$ for $i = 1, \ldots, n$. Let $h_0$ be the all-0 classifier. For a radius $r = 1/n$, the ball $B_H(h_0, r)$ contains any tree that misclassifies only one point. It is possible to construct a tree that isolates and flips the label of any single point $X_i$ (Figure 2b). Therefore, for any point $X_i$, there exists a hypothesis $h_i \in B_H(h_0, r)$ such that $h_i(X_i) = 1 \neq h_0(X_i)$. This implies that the entire dataset is in the disagreement region $\text{DIS}(B_H(h_0, r))$, yielding a coefficient of at least $\frac{|\text{DIS}|}{rn} = \frac{n}{\frac{1}{n} \cdot n} = n$.

### 3.4 RELAXING THE UNIFORMITY ASSUMPTION

The integer grid assumption for the input distribution is restrictive. We can relax it by assigning a weight $W_i \in [1, \lambda]$ to each data point $X_i$, representing its relative importance. This modifies the distance metric to a weighted average:

$$D_{S,W}(h_1, h_2) = \frac{\sum_{i=1}^{n} \mathbb{I}(h_1(X_i) \neq h_2(X_i))W_i}{\sum_{i=1}^{n} W_i}$$

This formulation generalizes the analysis of classification errors. As we prove in Theorem D.14 (which is a variant of Theorem 7.6 from Hanneke (2014) for discrete datasets), the disagreement coefficient for this weighted task is scaled by at most $\lambda^2$ compared to the unweighted case.

# 4    A MULTIPLICATIVE-ERROR-BOUND ACTIVE LEARNING ALGORITHM

This section introduces and analyzes an active learning algorithm designed to find a classifier with a multiplicative error guarantee. That is, if the optimal classifier $h^*$ in a class $H$ has an error of $\eta$, our algorithm returns a classifier $h$ with error less than $\eta(1 + \epsilon)$ with high probability.

A natural first question is whether existing algorithms, which typically provide additive error guarantees (i.e., returning $h$ with error at most $\eta + \epsilon'$), can be adapted for the multiplicative setting. However, such adaptations are fundamentally label-inefficient. Any approach based on an additive algorithm would require an estimate of the optimal error $\eta$ to set the additive term $\epsilon'$ appropriately (e.g., $\epsilon' = \epsilon\eta$). Estimating or verifying an error rate of $\eta$ with high probability requires $\Omega(1/\eta)$ samples. Since an effective algorithm's label complexity cannot depend on the unknown, and potentially very small, value of $\eta$, additive frameworks are unsuitable for achieving multiplicative guarantees. In Appendix E, we formally investigate the scenarios in which these adaptations fail, demonstrating their inherent limitations.

Our approach, by contrast, is designed to circumvent this dependence on $\eta$. We first present the algorithm's core logic in the simple, one-dimensional setting of a decision stump. We then generalize this framework to arbitrary binary classification tasks, yielding a result whose label complexity depends on the disagreement coefficient derived in Section 3.

## 4.1    THE DECISION STUMP CASE

A decision stump for one-dimensional data is a decision tree of depth one, defined by a single threshold. We begin with this setting to illustrate our algorithmic approach in a simple context. The main result for this section is the following label complexity bound.

**Theorem 4.1.** *For a one-dimensional, sorted dataset of size* $n$*, Algorithm 1 returns a* $(1 + \epsilon)$*-approximate decision stump with probability at least* $1 - \delta$*, using a total of*

$$O\left(\ln(n)\left(\ln(\ln(n)) + \ln\left(\frac{1}{\delta}\right)\right) + \frac{1}{\epsilon^2}\ln\left(\frac{1}{\delta\epsilon}\right)\right) \text{ label queries.}$$

**Problem Setup.**    Our approach operates within the active learning framework of Woodruff et al. Musco et al. (2022), where the algorithm has access to all input samples from the outset and can adaptively query their labels. Specifically, the algorithm is given a sorted vector of unlabeled data points $X \in \mathbb{R}^n$ ($X_i < X_{i+1}$) and can adaptively query their labels from the target vector $Y \in \{0, 1\}^n$. A stump classifier $h$ is defined by a threshold, which we can represent by the index of the first sample it classifies as 1. Thus, $h \in \{0, 1, \dots, n\}$ corresponds to the rule $h(x) = \mathbb{I}(x \geq X_h)$, where we define $X_0 = -\infty$ to handle the case where all samples are labeled 1. The error of a classifier $h$ on a subset of samples $S' \subseteq S$ is denoted $\text{err}_{S'}(h) := \frac{1}{|S'|}\sum_{(x,y)\in S'}\mathbb{I}(h(x) \neq y)$.

**Algorithm Intuition.**    Algorithm 1 maintains an interval of candidate stumps $[L_i, R_i]$ that, with high probability, contains the optimal one. Each iteration attempts to shrink this interval by (i) **sampling** a few labeled points from $X_{[L_i,R_i]}$, (ii) **bounding errors** of all classifiers using high-probability lower/upper bounds (Appendix A.1), and (iii) **pruning** any classifier $h'$ whose lower bound is above another's upper bound.

The crucial feature is how the algorithm reacts when pruning fails: if $[L_i, R_i]$ does not shrink by at least half, this signals that all classifiers in the interval incur relatively high error. Instead of wasting more iterations, the algorithm halts and directly estimates the best classifier in the range using $O(\frac{1}{\epsilon^2}\ln\frac{1}{\delta\epsilon})$ additional samples. We call this last phase *direct estimation* phase.

Formally, in iteration $i$ we obtain bounds for each $h$ using $O(\ln(1/\delta'))$ samples $S_i$, ensuring with probability $1 - \delta'$, where $\delta' = \delta/(2\log_2 2n)$ we have:

$$\text{LB}(S_i, h, \delta') \leq \text{err}_{[L_i, R_i]}(h) \leq \text{UB}(S_i, h, \delta'), \quad \text{UB} - \text{LB} \leq \tfrac{1}{16}.$$

Classifiers eliminated by these bounds shrink the interval; if the shrinkage is insufficient, the algorithm switches to *direct estimation*. Pseudocode is given in Algorithm 1. We write $\mathscr{R}(a, S)$ for a uniformly random subset of $S$ of size $a$; that is, we choose $a$ distinct elements from $S$ at random.

We empirically validated Algorithm 1 on datasets of size $n = 10^7$ with 0.1 label noise. The results demonstrate that the algorithm achieves success rates $> 90\%$ using constants significantly smaller

**Algorithm 1 Stump algorithm**

1: Initialize $i \leftarrow 0$
2: Initialize $L_i \leftarrow 0, R_i \leftarrow n$
3: **while** $L_i \leq R_i$ **do**
4: $\quad S_{i+1} \leftarrow \mathcal{R}(c_1 \ln(\frac{1}{\delta'}) + b_1, X_{[L_i, R_i]})$
5: $\quad i \leftarrow i + 1$
6: $\quad \beta \leftarrow \min\limits_{h \in [L_{i-1}, R_{i-1}]} \text{UB}(S_i, h, \delta')$
7: $\quad H \leftarrow \{h' \in [L_{i-1}, R_{i-1}] \mid \text{LB}(S_i, h', \delta') \leq \beta\}$
8: $\quad L_i \leftarrow \min(H), R_i \leftarrow \max(H)$
9: $\quad$ **if** $R_i - L_i > \frac{R_{i-1} - L_{i-1}}{2}$ **then**
10: $\quad\quad S' \leftarrow \mathcal{R}(\frac{c_2}{\epsilon^2}(\ln\frac{1}{\delta\epsilon} + b_2), X_{[L_{i-1}, R_{i-1}]})$
11: $\quad\quad$ Return $\arg\min\limits_{h \in [L_{i-1}, R_{i-1}]} \text{UB}(S', h, \frac{\delta}{2})$
12: $\quad$ **end if**
13: **end while**
14: Return $L_i$

**Algorithm 2 General Binary Classification**

1: $S \leftarrow$ All samples, $H \leftarrow$ All classifiers
2: $\theta \leftarrow$ Calculate $\theta$ for using Definition 3.3.
3: $i \leftarrow 0$
4: $H_i \leftarrow H, r_i \leftarrow 1$ {Initial progress measure}
5: **while** $|H_i| > 1$ **do**
6: $\quad S_i \leftarrow \mathcal{R}(c_1 \theta^2(V_H \ln\theta + \ln\frac{1}{\delta'}) + b_1, \text{DIS}(H_i))$
7: $\quad \beta \leftarrow \min_{h \in H_i} \text{UB}(S_i, h, \delta')$
8: $\quad H_{i+1} \leftarrow \{h \in H_i \mid \text{LB}(S_i, h, \delta') \leq \beta\}$
9: $\quad r_{i+1} \leftarrow \text{radius}(H_{i+1})$
10: $\quad$ **if** $r_{i+1} > \frac{r_i}{2}$ **then**
11: $\quad\quad S' \leftarrow \mathcal{R}(\frac{c_2 \theta^2}{\epsilon^2}(V_H \ln(\frac{\theta}{\epsilon}) + \ln(\frac{1}{\delta})) + b_2, \text{DIS}(H_i))$
12: $\quad\quad$ Return $\arg\min_{h \in H_i} \text{UB}(S', h, \frac{\delta}{2})$
13: $\quad$ **end if**
14: $\quad i \leftarrow i + 1$
15: **end while**
16: Return $h \in H$

than the theoretical worst-case ($c_1, b_1 \approx 3$), supporting its practical viability. Full details are in Appendix F.

**Theorem 4.2.** *There exist universal constants $c_1, c_2, b_1, b_2$ such that Algorithm 1 returns a classifier with an error rate less than $\eta(1 + \epsilon)$ with probability at least $1 - \delta$ when provided with a one-dimensional dataset.*

**Correctness and Label Complexity Proof Sketch.** The proof proceeds in several steps. First, Lemma B.1 shows that the main loop of Algorithm 1 executes at most $log_2 2n$ iterations. Then, Lemma B.2 establishes that, with probability at least $1 - delta$, all bounds produced by the algorithm hold simultaneously throughout its execution. Conditioning on this event, we next prove that the optimal classifier is never eliminated. The argument is as follows: if a classifier $h$ is suboptimal within the interval $[L_i, R_i]$, then $h$ cannot be optimal over the entire dataset, as shown in Lemma B.3. Combining this with Lemma B.4, we conclude that the optimal classifier always remains in the candidate range $[L_i, R_i]$. Consequently, when the algorithm terminates, the returned classifier is guaranteed to be optimal among the remaining candidates.

The main subtlety arises from the two different ways the algorithm can terminate: by continuing to shrink intervals, or by entering the *direct estimate* phase. The key idea is that these two outcomes correspond to complementary regimes for the error of the optimal classifier. When the optimal classifier has small error on the current interval $[L_i, R_i]$, Lemma B.6 shows that the interval length shrinks rapidly. In fact, if the optimal error is less than $\frac{1}{16}$, then the next interval $[L_{i+1}, R_{i+1}]$ is at most half the size of $[L_i, R_i]$. Thus, in the low-error regime the algorithm never enters the *direct estimate* phase; instead, it keeps shrinking intervals until the candidate set is tightly localized around the optimum. In contrast, if the algorithm does enter the *direct estimate* phase, this indicates that the optimal error on the current interval is relatively large. In this high-error regime, approximating within a factor of $(1 + \epsilon)$ becomes easier. Lemma B.7 formalizes this intuition, showing that there exist universal constants $c_2$ and $b_2$ such that the classifier returned in the direct estimate phase always achieves error within the desired $(1 + \epsilon)$-factor guarantee. Putting the cases together, we conclude that the algorithm always returns a $(1 + \epsilon)$-approximate classifier with probability at least $1 - \delta$, thereby proving Theorem 4.2.

To establish the label complexity bound in Theorem 4.1, we first apply Lemma B.1 to show that the for loop repeats at most $log_2 2n$ times. During these iterations, the algorithm uses at most $O(\ln n \ln \frac{\ln n}{\delta})$ label queries. If the algorithm later enters the *direct estimate* phase, it performs an additional $O\left(\frac{1}{\epsilon^2} \ln \frac{1}{\delta\epsilon}\right)$ label queries. This completes the proof of Theorem 4.1.

## 4.2 Lower Bound

We aim to demonstrate that any active learning algorithm within the given setting has a label complexity of $\Omega(\ln(\frac{1}{\delta}) \cdot \frac{1}{\epsilon^2})$. This result establishes that it is not possible to significantly improve the label complexity with respect to the term $\epsilon$, beyond a logarithmic factor. The result is as follows:

**Theorem 4.3.** *Any active learning algorithm requires* $\Omega\left(\ln\left(\frac{1}{\delta}\right)\frac{1}{\epsilon^2}\right)$ *queries to return a* $(1 + \epsilon)$-*approximate decision stump with probability greater than* $1 - \delta$.

To prove Theorem 4.3, we build upon the lower bound established in Kääriäinen (2006), which determines the minimum number of coin tosses required to decide whether heads is more likely than tails. We adapt this result by modeling the active learning problem as an analogous coin-tossing process: here, the "coin" provides the requested labels, and the active learning algorithm's classifier determines whether heads is more likely than tails. By applying the lower bound from Kääriäinen (2006) to this framework, we derive a lower bound for the label complexity of the active learning algorithm. We should mention that this result were proven for continues input spaces Hanneke (2014) but needed additional techniques for discrete datasets.

## 4.3 Generalization to Arbitrary Classifiers

We now generalize the stump algorithm to handle any binary classification task. The performance of this resulting algorithm is formally stated in Theorem 1.2 in the introduction. The resulting algorithm's performance depends on structural properties of the hypothesis class, captured by the VC dimension and the disagreement coefficient.

Algorithm 2 follows the same template as the stump algorithm but replaces 1D-specific concepts with their general counterparts. The candidate set is not an interval $[L_i, R_i]$ but a general subset of hypotheses $H_i \subseteq H$. The sampling occurs not from a data range but from the **disagreement region**, $\text{DIS}_S(H_i)$. This is the set of points informative for distinguishing among remaining classifiers. Finally, progress is measured not by interval length but by the **radius** of the hypothesis set, radius($H_i$), which is the radius of the smallest ball containing $H_i$. The pruning step remains the same: eliminate classifiers that are provably worse than another candidate. The algorithm switches to a final, direct estimation phase if the radius of the hypothesis set fails to halve in an iteration.

**Role of the Disagreement Coefficient.** The proof of Theorem 4.4 shows that an ineffective pruning step implies a high optimal error. In this general setting, "high" is relative to the disagreement coefficient. Specifically, if the radius fails to halve, the optimal error $\eta_i$ on the disagreement region must be at least $\Omega(1/\theta)$. The disagreement coefficient $\theta$ bridges the gap between the radius of the hypothesis ball and the size of the disagreement region, allowing us to make this critical inference.

**Algorithmic Details.** Algorithm 2 formalizes this procedure. The core of the algorithm is an iterative loop that prunes the set of candidate classifiers, $H_i$. In iteration $i$, we focus on samples $x \in S$ for which there exist $h_1, h_2 \in H_i$ such that the two classifiers disagree with each other $h_1(x) \neq h_2(x)$ or more formally $\text{DIS}(H_i)$. Therefore, in each iteration, a sample set $S_i$ is drawn from the disagreement region $\text{DIS}(H_i)$ (Line 6). Using this sample, the algorithm finds the minimum error upper bound $\beta$ among all classifiers in $H_i$ and then forms the next set, $H_{i+1}$, by eliminating any hypothesis whose error lower bound exceeds $\beta$ (Lines 7-8). This efficiently removes classifiers that are provably suboptimal based on the evidence from $S_i$.

**Measuring Progress.** Progress is tracked via the **radius** of the hypothesis set, which quantifies its size. The radius of a hypothesis set $H_i$ is then the radius of the smallest ball, under this metric, that encloses all classifiers in the set:

$$\text{radius}(H_i) := \min\{r \mid \exists_{h' \in H_i}, H_i \subseteq B_{D_{S'}}(h', r)\}.$$

Ff an iteration fails to halve this radius ($r_i > r_{i-1}/2$), the algorithm transitions to its final estimation phase (Lines 10-12). This switch is justified because slow progress implies a high optimal error, which allows us to select the final classifier. The correctness of the general algorithm is formally stated below.

**Theorem 4.4.** *There exist universal constants* $c_1, c_2, b_1, b_2$ *such that, for any binary classification task, Algorithm 2 returns a classifier with error less than* $\eta(1 + \epsilon)$ *with probability at least* $1 - \delta$, *where* $\eta$ *is the error of the optimal classifier.*

The proof mirrors the stump case. We show the main loop runs only $O(\log n)$ times (Lemma C.1) and that all probabilistic bounds hold simultaneously with high probability (Lemma B.2). Crucially, the optimal classifier $h^*$ is never pruned from $H_i$ (Lemma C.5). The argument ties progress to the optimal error via the disagreement coefficient. Lemma C.7 shows that if the optimal error on $\text{DIS}(H_i)$ is small (below $1/(16\theta)$), the radius must halve. Otherwise, Lemma C.8 ensures the *direct estimate* phase suffices to output a $(1 + \epsilon)$-approximate classifier.

Finally, Corollary 1.3 follows directly from Theorem 1.1 and Theorem 1.2, with details in Appendix D.1.

## 5 CONCLUSION

In this paper, we established the first rigorous theoretical foundation for actively learning decision trees, presenting an algorithm that achieves a polylogarithmic label complexity in the dataset size. This result is built on two core innovations: the first analysis of the disagreement coefficient for decision trees, which we bound as $\theta = O(\ln^d(n))$, and a proof that our underlying assumptions—unique feature dimensions per path and a grid-like data structure—are necessary to avoid polynomial complexity. We combined this with the introduction of the first general active learning algorithm for any binary classification task to provide a $(1 + \epsilon)$-multiplicative error guarantee, a more robust framework than traditional additive models whose dependence on $\epsilon$ we show is nearly optimal. Our work bridges a critical gap between the practical use of decision trees and their theoretical understanding, opening several avenues for future research, such as relaxing our structural assumptions, extending the analysis to continuous data domains, and applying our general algorithmic framework to other classifier classes.

### ACKNOWLEDGMENTS

This work is partially supported by DARPA expMath, ONR MURI 2024 award on Algorithms, Learning, and Game Theory, Army-Research Laboratory (ARL) grant W911NF2410052, NSF AF:Small grants 2218678, 2114269, 2347322.

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

## APPENDIX OUTLINE

In Appendix A, we explain how to calculate the lower and upper bounds, LB and UB, in general. Next, in Appendix A.1, we specifically determine these bounds for stumps, and in Appendix A.2, we determine them for decision trees.

Then, we present the proofs of the theorems and lemmas from the main body, starting with the proofs of Section 4.1 in Appendix B. Specifically, we first provide some required lemmas. In Appendix B.1, we then provide proofs for Theorem 4.2, which proves that Algorithm 1 is correct. Later, in Appendix B.2, we prove Theorem 4.1, which determines the label complexity of Algorithm 1. Finally, in Appendix B.3, we prove Theorem 4.3, which provides a lower bound on the label complexity of any active learning algorithm.

In Appendix C, we provide the proofs for the theorems in Section 4.3. We first introduce and prove some required lemmas, which serve as the foundation for proving Theorem 4.4 and Theorem 1.2. These results establish the correctness of Algorithm 2 and analyze its label complexity, respectively.

In Appendix D, we provide proofs for the lemmas introduced in Section 3. After proving these lemmas, we proceed to Theorem 1.1 and Proposition 3.5, which calculate the disagreement coefficient for decision trees and line trees, respectively. In Appendix D.1, we present the proof of our main result, Corollary 1.3, which calculated the label complexity of our algorithm for a decision tree. Additionally, in Appendix D.3, we provide proofs of the necessity of our assumptions, i.e., Theorem 3.6 and Theorem 3.7. Finally, in Appendix D.4, we show how you can partially relax the uniformity assumption, though not remove it entirely.

In Appendix E, we explain why additive algorithms fail in a multiplicative setting and explain the relation between additive and multiplicative settings and algorithms.

In Appendix F, we present an empirical analysis of our **Stump** algorithm, showing that, in practice, small values for $c_1, c_2, b_1$, and $b_2$ can be chosen while still ensuring the algorithm performs as intended.

TABLE OF NOTATION

For convenience, we provide a summary of the key mathematical notations used throughout the algorithms and analysis in Table 1.

| Symbol | Description |
|---|---|
| $S, n$ | The dataset $S$ and its size $n$. |
| $H$ | The hypothesis class (e.g., decision trees). |
| $h^*$ | The optimal classifier in $H$ (minimizes classification error). |
| $\mathcal{R}(m, S)$ | A uniformly random subset of $S$ of size $m$ (sampling without replacement). |
| $err_S(h)$ | The empirical error of classifier $h$ on set $S$: $\frac{1}{|S|} \sum_{(x,y)\in S} \mathbb{I}(h(x) \neq y)$. |
| $D_S(h, h')$ | The distance between two classifiers: fraction of points in $S$ where they disagree. |
| $B_H(h, r)$ | The ball of classifiers in $H$ within distance $r$ of $h$. |
| $\text{DIS}_S(V)$ | The disagreement region: points in $S$ where at least two classifiers in $V$ disagree. |
| $\theta$ | The disagreement coefficient. |
| $V_H$ | The VC dimension of the hypothesis class $H$. |
| $\text{LB}(S, h, \delta)$ | Lower bound on the error of $h$ with confidence $1 - \delta$. |
| $\text{UB}(S, h, \delta)$ | Upper bound on the error of $h$ with confidence $1 - \delta$. |

Table 1: Summary of notation used in Algorithms 1 and 2

A    CALCULATION OF ERROR BOUNDS

In this section, we formally define the error metrics used to derive our bounds. For a hypothesis $h$ and a target dataset (or subset) $S$, we denote the true error as $err_S(h)$, representing the fraction of misclassified points in $S$:

$$err_S(h) = \frac{1}{|S|} \sum_{(x,y)\in S} \mathbb{I}(h(x) \neq y)$$

When the algorithm draws a random subsample $S' \subseteq S$, we calculate the empirical error $\hat{err}(h)$ restricted to this sample:

$$\hat{err}(h) = \frac{1}{|S'|} \sum_{(x,y)\in S'} \mathbb{I}(h(x) \neq y)$$

Our algorithms do not use $\hat{err}(h)$ directly for pruning; rather, they rely on high-probability Lower Bounds (LB) and Upper Bounds (UB) derived from $\hat{err}(h)$ to estimate the true error $err_S(h)$.

To compute the lower and upper bounds (LB and UB), we leverage the following theorem from Anthony & Bartlett (2002); Balcan et al. (2006):

**Theorem A.1.** *Let $H$ be a hypothesis class of functions mapping from $X$ to $\{-1, 1\}$, with a finite VC-dimension $V_H \geq 1$. Let $D$ be an arbitrary, but fixed, probability distribution over $X \times \{-1, 1\}$. For any $\epsilon, \delta > 0$, if a sample is drawn from $D$ with size*

$$m(\epsilon, \delta, V_H) = \frac{64}{\epsilon^2} \left( 2V_H \ln \left( \frac{12}{\epsilon} \right) + \ln \left( \frac{4}{\delta} \right) \right),$$

*then, with probability at least $1 - \delta$, the following holds for all $h \in H$:*

$$|err(h) - \hat{err}(h)| \leq \epsilon.$$

Using Theorem A.1, we can derive the error bounds. From the theorem, we know that $|\text{err}(h) - \hat{\text{err}}(h)| \leq \epsilon$. Consequently, we can define the lower and upper bounds as follows:

$$\text{LB}(h) = \hat{\text{err}}(h) - \epsilon \quad \text{and} \quad \text{UB}(h) = \hat{\text{err}}(h) + \epsilon.$$

These bounds are valid with probability greater than $1 - \delta$.

## A.1 STUMP

For decision stumps, the VC-dimension $V_H$ is known to be 1, as established in Lemma A.2. Hence, applying the formula for sample size in Theorem A.1, we obtain the following sample size requirement:

$$m(\epsilon, \delta) = \frac{256}{\epsilon^2} \left( 2 \ln \left( \frac{24}{\epsilon} \right) + \ln \left( \frac{4}{\delta} \right) \right).$$

After sampling, we can compute the empirical error for each classifier. Subsequently, we define the lower and upper bounds for each classifier as:

$$\text{LB}(h) = \hat{\text{err}}(h) - \frac{\epsilon}{2} \quad \text{and} \quad \text{UB}(h) = \hat{\text{err}}(h) + \frac{\epsilon}{2}.$$

This guarantees, with probability at least $1 - \delta$, that for all hypotheses $h$, the following holds:

$$\text{LB}(h) \leq \text{err}(h) \leq \text{UB}(h),$$

and additionally, the width of the bounds is exactly $\epsilon$, i.e., $\text{UB}(h) - \text{LB}(h) = \epsilon$.

**Lemma A.2.** *Let $\mathcal{H}$ denote the hypothesis class of decision stumps in one dimension, where each hypothesis $h \in \mathcal{H}$ assigns the label 1 to points exceeding a threshold $\theta \in \mathbb{R}$ and 0 otherwise. Then, the VC-dimension of $\mathcal{H}$ is 1.*

*Proof.* To prove that the VC-dimension of $\mathcal{H}$ is 1, we must show that there exists a set of one point that can be shattered by $\mathcal{H}$, but no set of two points can be shattered.

**Shattering a single point:** Consider a single point $x_1 \in \mathbb{R}$. By choosing an appropriate threshold $\theta$, we can label $x_1$ as either 0 or 1. Thus, a set of one point can be shattered by $\mathcal{H}$.

**Inability to shatter two points:** Now, consider a set of two points, $x_1, x_2 \in \mathbb{R}$ with $x_1 < x_2$. There are four possible labelings: $(0,0), (0,1), (1,0), (1,1)$. However, the labeling $(1,0)$ cannot be achieved. If we choose a threshold $\theta$ such that $x_1 < \theta < x_2$, we obtain the labeling $(0,1)$. If we choose $\theta \leq x_1$, we get $(0,0)$, and if we choose $\theta \geq x_2$, we obtain $(1,1)$. Since $(1,0)$ is impossible, the set $\{x_1, x_2\}$ cannot be shattered. Therefore, no set of two points can be shattered.

Since there exists a set of one point that can be shattered, but no set of two points can be, the VC-dimension of $\mathcal{H}$ is 1. $\qquad\square$

## A.2 DECISION TREE

The VC-dimension, $V_H$, of decision trees with height at most $d$ in dim-dimensional data is $O\left(2^d(d + \ln \text{dim})\right)$, as established in Leboeuf et al. (2020). Utilizing this, along with Theorem A.1, we can achieve error bounds where $\text{UB}(h) - \text{LB}(h) \leq \epsilon$ with the following number of samples:

$$O\left( \frac{1}{\epsilon^2} \left( 2^d(d + \ln \text{dim}) \ln \frac{1}{\epsilon} + \ln \frac{1}{\delta} \right) \right).$$

More specifically, using Lemma A.3 and Theorem A.1, we require:

$$\frac{256}{\epsilon^2} \left( 20 \cdot 2^d(d + \log_2 \text{dim}) \ln \left( \frac{24}{\epsilon} \right) + \ln \left( \frac{4}{\delta} \right) \right)$$

samples.

**Lemma A.3.** *Let $H$ be a decision tree of height at most $d$, where each node uses one of $\text{dim} \geq 2$ data dimensions. The VC dimension of $H$, $V_H$, is at most $10 \cdot 2^d(d + \log_2 \text{dim})$.*

*Proof of Lemma A.3.* Based on Leboeuf et al. (2020), the VC dimension $V_H$ satisfies $V_H \leq \max\{m \mid (14m \cdot \dim)^N \geq 2^m\}$, where dim is the number of dimensions and $N$ is the number of internal nodes. For a height $d$, $N \leq 2^d - 1$. Thus, $V_H \leq \max\{m \mid (14m \cdot \dim)^{2^d-1} \geq 2^m\}$. If $(14m \cdot \dim)^{2^d} \geq 2^m$, we simplify by assuming $m = 2^d(d + \log_2 \dim)c$ and get $14\dim \cdot 2^d(d + \log_2 \dim)c \geq 2^{(d+\log_2 \dim)c}$. Dividing by $2^d$, we derive:

$$14\dim(d + \log_2 \dim)c \geq 2^{d(c-1)}\dim^c.$$

For $c = 10$, this inequality fails, since

$$140\dim(d + \log_2 \dim) \geq 2^{h \cdot (c-1)}\dim^c = 2^{d \cdot (c-1)}\dim + 2^{d \cdot (c-1)}(\dim^c - \dim).$$

Here, $\log_2 d < d$ and $\log_2(140) < 8 \leq 8d \leq (c-2)d$, implying $\log_2(140) + \log_2 d < d \cdot (c - 1)$. Therefore, $140d < 2^{d \cdot (c-1)}$. Adding dim to both sides gives:

$$140d \cdot \dim < 2^{d \cdot (c-1)}\dim.$$

Since $\dim \geq 2$, and $\log_2 \dim < \dim$ we get: $\log_2 \dim < \dim^{(c-2)}(\dim - 1)$. Adding $140\dim$ to both sides and noting $d \leq \dim$ we get

$$140\dim \log_2 \dim < 140\dim^{c-1}(d - 1) < 2^9\dim^{c-1}(\dim - 1) \leq 2^{d \cdot 9}\dim^{c-1}(\dim - 1) \leq$$
$$2^{d \cdot (c-1)}\dim^{c-1}(\dim - 1) \leq 2^{d \cdot (c-1)}\dim^c.$$

Hence:

$$14\dim \log_2 \dim \cdot c < 2^{d \cdot (c-1)}\dim^c \Rightarrow V_H \leq 10 \cdot 2^d(d + \log_2 \dim).$$

$\square$

# B   STUMP PROOFS

In this section, we present the proofs associated with Section 4.1, which pertain to Algorithm 1. These include proofs of its correctness and label complexity. Additionally, we establish a lower bound for the label complexity of any active learning algorithm.

We begin by proving that the loop in Algorithm 1 repeats at most $\log_2(2n)$ times.

**Lemma B.1.** *In the execution of Algorithm 1, we enter the loop at most $\log_2(n)$ times.*

*Proof of Lemma B.1.* At the start of the algorithm, we have $R_i - L_i = n$, and this value is halved in each iteration. When it reaches 1, the algorithm terminates—either by entering the *If statement* or by reducing $R_i - L_i$ to 0. Therefore, the number of iterations is at most $\log_2(2n)$. $\square$

To prove Theorem 4.2, we need to establish that all lower and upper bounds are valid during the algorithm's execution simultaneously with probability at least $1 - \delta$.

**Lemma B.2.** *In an execution of Algorithm 1, all estimated lower and upper bounds are valid with probability at least $1 - \delta$.*

*Proof of Lemma B.2.* From Lemma B.1, we know that the bounds outside the *If statement* are evaluated at most $\log_2(2n)$ times. Each time, the bounds are correct with probability at least $1 - \delta'$. Thus, the probability of a bound being incorrect is less than $\delta'$. Therefore, the probability of at least one bound being incorrect is less than $\delta' \log_2(2n)$. Using the definition of $\delta'$, we know $\delta' = \frac{\delta}{2 \log_2(2n)}$, so we have $\delta' \log_2(2n) = \frac{\delta}{2}$.

Additionally, the bounds inside the *If statement* are evaluated only once, and the probability of them being wrong is less than $\frac{1}{2\delta}$. Combining these two factors, the overall probability of any bound being incorrect is less than $\delta$. $\square$

Next, we must prove that the optimal classifier is never eliminated if all bounds are valid. This is formalized in Lemma B.4, which builds upon Lemma B.3. Since we estimate the error using random samples from the disagreement set rather than the entire dataset, we must first relate the error

of classifiers in the disagreement set to their overall error. Lemma B.3 establishes that the error relationship between two classifiers remains consistent across the disagreement set and the total dataset if all the samples on which they disagree are contained within the disagreement set. Specifically, if one classifier has a larger error within the disagreement set, it will also have a larger error on the total dataset. Consequently, the optimal classifier within the disagreement set is guaranteed to be the overall optimal classifier.

**Lemma B.3.** *For two classifiers $h_1$ and $h_2$, if we have $err_S(h_1) \leq err_S(h_2)$, then for any subset of samples $S'$ containing all samples where $h_1$ and $h_2$ disagree, i.e., $\{x \in S \mid h_1(x) \neq h_2(x)\} \subseteq S'$, we will have $err_{S'}(h_1) \leq err_{S'}(h_2)$.*

*Proof of Lemma B.3.* Applying definition of err, to our assumption $\text{err}_S(h_1) \leq \text{err}_S(h_2)$, we have:

$$\sum_{(x,y)\in S} \mathbb{I}(h_1(x) \neq y) \leq \sum_{(x,y)\in S} \mathbb{I}(h_2(x) \neq y)$$

Now, splitting $S$ into $S'$ and its complement $S \setminus S'$:

$$\sum_{(x,y)\in S'} \mathbb{I}(h_1(x) \neq y) + \sum_{(x,y)\in S\setminus S'} \mathbb{I}(h_1(x) \neq y) \leq \sum_{(x,y)\in S'} \mathbb{I}(h_2(x) \neq y) + \sum_{(x,y)\in S\setminus S'} \mathbb{I}(h_2(x) \neq y)$$

Since $S'$ contains all the samples where $h_1$ and $h_2$ disagree, the error on $S \setminus S'$ will be identical for both classifiers. Therefore:

$$\sum_{(x,y)\in S'} \mathbb{I}(h_1(x) \neq y) \leq \sum_{(x,y)\in S'} \mathbb{I}(h_2(x) \neq y)$$

Hence: $\text{err}_{S'}(h_1) \leq \text{err}_{S'}(h_2)$ □

**Lemma B.4.** *If all bounds in the execution of Algorithm 1 are valid, the algorithm will never eliminate any optimal classifiers.*

*Proof of Lemma B.4.* We prove by contradiction. Suppose that there exists an iteration $i$ in which an optimal classifier $h^*$ is eliminated, while all bounds are valid in that step. This implies that $h^* \in [L_{i-1}, R_{i-1}]$ but $h^* \notin [L_i, R_i]$. Thus, we have:

$$\text{LB}(S_i, h^*, \delta') > \min_{h\in[L_{i-1},R_{i-1}]} \text{UB}(S_i, h, \delta')$$

Let $h'$ be the classifier such that: $\min_{h\in[L_{i-1},R_{i-1}]} \text{UB}(S_i, h, \delta') = \text{UB}(S_i, h', \delta')$ Then: $\text{LB}(S_i, h^*, \delta') > \text{UB}(S_i, h', \delta')$.

Let $S'$ be the set of all samples in the range of valid classifiers, i.e., $S' = \{X_{[L_{i-1},R_{i-1}]}\}$. Since the bounds are assumed to be correct, we know:

$$\text{err}_{S'}(h^*) \geq \text{LB}(S_i, h^*, \delta') > \text{UB}(S_i, h', \delta') \geq \text{err}_{S'}(h') \tag{2}$$

Since both $h^*$ and $h'$ belong to the interval $[L_{i-1}, R_{i-1}]$, we know that $[L_{i-1}, R_{i-1}]$ contains all the samples where $h^*$ and $h'$ disagree. Given that $h^*$ is the optimal classifier, we have: $\text{err}_S(h^*) \leq \text{err}_S(h')$.

Using Lemma B.3, we conclude: $\text{err}_{S'}(h^*) \leq \text{err}_{S'}(h')$. This contradicts Inequality 2. Thus, $h^*$ cannot be eliminated. □

We now aim to demonstrate that classifiers far from $h^*$ have high error rates on $X_{[L_i,R_i]}$, ensuring they will be eliminated by $h^*$. This claim is formally established in Lemma B.5 below.

**Lemma B.5.** *For all $h \in [L_i, R_i]$, in any iteration $i$, the following inequality holds:*

$$err_{X_{[L_i,R_i]}}(h) \geq \frac{|h - h^*|}{R_i - L_i + 1} - err_{X_{[L_i,R_i]}}(h^*),$$

*Proof of Lemma B.5.* We begin by recalling the definition of the error function $\text{err}_{X_{[L_i,R_i]}}(h)$:

$$\text{err}_{X_{[L_i,R_i]}}(h) = \frac{1}{R_i - L_i + 1} \sum_{j \in [L_i,R_i]} \mathbb{I}(h(X_j) \neq Y_j),$$

Without loss of generality, assume that $h < h^*$. This assumption allows us to focus on the data points where $h$ and $h^*$ make different predictions. $h$ and $h^*$ differ in their predictions on samples $X_j$, where $h \leq j < h^*$. Also, let $M$ denote the number of misclassifications made by both $h$ and $h^*$ outside the range $[h, h^*)$, but in $[L_i, R_i]$. Since $h$ and $h^*$ behave identically on samples outside the range $[h, h^*)$, their misclassifications outside of the range are equivalent.

Thus, the error of $h$ can be expressed as:

$$\text{err}_{X_{[L_i,R_i]}}(h) = \frac{1}{R_i - L_i + 1} \left( \sum_{j \in [h,h^*)} \mathbb{I}(h(X_j) \neq Y_j) + M \right).$$

Because $h$ and $h^*$ make different predictions on $X_j$ for $j \in [h, h^*)$, we have:

$$\text{err}_{X_{[L_i,R_i]}}(h) = \frac{1}{R_i - L_i + 1} \left( \left( \sum_{j \in [h,h^*)} 1 - \mathbb{I}(h^*(X_j) \neq Y_j) \right) + M \right).$$

Here, $\sum_{j \in [h,h^*)} 1$ counts the total number of samples in $X_{[h,h^*)}$, while $\sum_{j \in [h,h^*)} \mathbb{I}(h^*(X_j) \neq Y_j)$ counts the number of misclassifications made by $h^*$. Simplifying further:

$$\text{err}_{X_{[L_i,R_i]}}(h) = \frac{|h - h^*|}{R_i - L_i + 1} - \frac{1}{R_i - L_i + 1} \left( \sum_{j \in [h,h^*)} \mathbb{I}(h^*(X_j) \neq Y_j) + M \right) + \frac{2M}{R_i - L_i + 1}. \tag{3}$$

Given that $M$ represents the number of misclassifications made by $h^*$ outside $[h, h^*)$, the error of $h^*$ can be expressed as:

$$\text{err}_{X_{[L_i,R_i]}}(h^*) = \frac{1}{R_i - L_i + 1} \left( \sum_{j \in [h,h^*)} \mathbb{I}(h^*(X_j) \neq Y_j) + M \right).$$

Substituting this into Equality 3:

$$\text{err}_{X_{[L_i,R_i]}}(h) = \frac{|h - h^*|}{R_i - L_i + 1} - \text{err}_{X_{[L_i,R_i]}}(h^*) + \frac{2M}{R_i - L_i + 1}.$$

Since $M > 0$, it follows that:

$$\text{err}_{X_{[L_i,R_i]}}(h) \geq \frac{|h - h^*|}{R_i - L_i + 1} - \text{err}_{X_{[L_i,R_i]}}(h^*).$$

This completes the proof of Lemma B.5. $\qquad\square$

We now prove that if the optimal classifier's error is sufficiently low on $X_{[L_{i-1},R_{i-1}]}$, the algorithm can successfully reduce the range $[L_{i-1}, R_{i-1}]$ to half its size in $[L_i, R_i]$.

**Lemma B.6.** *There exist universal constants $c_1$ and $b_1$ such that in Algorithm 1, if at some iteration $i$, $\text{err}_{X_{[L_{i-1},R_{i-1}]}}(h^*) \leq \frac{1}{16}$, then we have:*

$$R_i - L_i \leq \frac{R_{i-1} - L_{i-1}}{2},$$

*provided all lower and upper bounds are valid during the algorithm's execution.*

*Proof of Lemma B.6.* We aim to demonstrate that all classifiers with a distance greater than $\frac{R_{i-1}-L_{i-1}}{4}$ from $h^*$ will be eliminated by $h^*$ itself.

Define $S'$ as the set of all samples within the range of remaining classifiers in iteration $i-1$, so $S' = X_{[L_{i-1}, R_{i-1}]}$.

Given that $\text{err}_{S'}(h^*) \leq \frac{1}{16}$ and by examining labels of $c_1 \ln \frac{1}{\delta'} + b_1$ samples, it follows from Appendix A.1 that for all $h$,

$$\text{LB}(S_i, h, \delta') \leq \text{err}_{S_i}(h) \leq \text{UB}(S_i, h, \delta'), \text{ and } \text{UB}(S_i, h, \delta') - \text{LB}(S_i, h, \delta') < \frac{1}{16},$$

provided that

$$\frac{256}{\left(\frac{1}{16}\right)^2}(2\ln(24) + \ln(\frac{4}{\delta'})) \leq c_1 \ln \frac{1}{\delta'} + b_1.$$

which will be satisfied by large enough $c_1$ and $b_1$. Thus, for $h^*$, we have:

$$\text{UB}(S_i, h^*, \delta') < \text{err}_{S'}(h^*) + \frac{1}{16} \leq \frac{1}{16} + \frac{1}{16} = \frac{1}{8}.$$

Using Lemma B.5, it follows:

$$\text{err}_{S'}(h) \geq \frac{|h - h^*|}{R_{i-1} - L_{i-1} + 1} - \text{err}_{S'}(h^*) > \frac{1}{4} - \frac{1}{16} = \frac{3}{16}.$$

Therefore,

$$\text{LB}(S_i, h, \delta') > \frac{3}{16} - \frac{1}{16} = \frac{1}{8}.$$

Hence, all classifiers $h$ such that $\frac{R_{i-1}-L_{i-1}+1}{4} \leq |h - h^*|$ will be eliminated. From this, we determine that $R_i = \max(H_i) < h^* + \frac{R_{i-1}-L_{i-1}+1}{4}$ and $L_i = \min(H_i) > h^* - \frac{R_{i-1}-L_{i-1}+1}{4}$. Therefore, $R_i - L_i \leq \frac{R_{i-1}-L_{i-1}}{2}$.

$\square$

Next, we demonstrate that if the algorithm enters the *If statement* and the optimal classifier has a high error in the disagreement range $[L_{i-1}, R_{i-1}]$, the algorithm will produce a sufficiently accurate classifier.

**Lemma B.7.** *There exist universal constants $c_2$ and $b_2$ such that if, in Algorithm 1, we have*

$$\text{err}_{X_{[L_{i-1}, R_{i-1}]}}(h^*) > \frac{1}{16},$$

*and the algorithm enters the If statement, It will return a classifier like $h$ with*

$$\text{err}_S(h) \leq (1 + \epsilon)\text{err}_S(h^*),$$

*provided that all lower and upper bounds are valid during the algorithm's execution.*

*Proof of Lemma B.7.* When Algorithm 1 enters the *If statement*, it constructs the set $S'$, comprising:

$$\frac{c_2}{\epsilon^2}\left(\ln\left(\frac{1}{\delta\epsilon}\right) + b_2\right)$$

random samples drawn from the interval $X_{[L_{i-1}, R_{i-1}]}$.

Let $h'$ denote the classifier returned by the algorithm, i.e.,

$$h' = \arg\min_{h \in [L_i, R_i]} \text{UB}(S', h, \frac{\delta}{2}).$$

From Appendix A.1, with sufficiently large $c_2$ and $b_2$, it follows that for all $h \in [L_i, R_i]$:

$$\text{UB}(S', h, \frac{\delta}{2}) \leq \text{err}_{X_{[L_{i-1}, R_{i-1}]}}(h) + \frac{\epsilon}{16}.$$

Since $h'$ is chosen to minimize the upper bound, we know: $\text{UB}(S', h', \frac{\delta}{2}) \leq \text{UB}(S', h^*, \frac{\delta}{2})$. Thus:

$$\text{err}_{X_{[L_{i-1}, R_{i-1}]}}(h') \leq \text{UB}(S', h', \frac{\delta}{2}) \leq \text{UB}(S', h^*, \frac{\delta}{2}) \leq \text{err}_{X_{[L_{i-1}, R_{i-1}]}}(h^*) + \frac{\epsilon}{16}.$$

From the definition of the error metric err, we write:

$$\frac{1}{R_{i-1} - L_{i-1} + 1} \sum_{j \in [L_{i-1}, R_{i-1}]} \mathbb{I}(h'(X_j) \neq Y_j) \leq \frac{1}{R_{i-1} - L_{i-1} + 1} \sum_{j \in [L_{i-1}, R_{i-1}]} \mathbb{I}(h^*(X_j) \neq Y_j) + \frac{\epsilon}{16}.$$

Multiplying through by $R_{i-1} - L_{i-1} + 1$, we obtain:

$$\sum_{j \in [L_{i-1}, R_{i-1}]} \mathbb{I}(h'(X_j) \neq Y_j) \leq \sum_{j \in [L_{i-1}, R_{i-1}]} \mathbb{I}(h^*(X_j) \neq Y_j) + \frac{(R_{i-1} - L_{i-1} + 1)\epsilon}{16}.$$

Outside the interval $[L_{i-1}, R_{i-1}]$, $h'$ and $h^*$ behave identically. Let $M$ denote the number of samples they misclassify outside the interval $[L_{i-1}, R_{i-1}]$. Adding $M$ to both sides:

$$M + \sum_{j \in [L_{i-1}, R_{i-1}]} \mathbb{I}(h'(X_j) \neq Y_j) \leq M + \sum_{j \in [L_{i-1}, R_{i-1}]} \mathbb{I}(h^*(X_j) \neq Y_j) + \frac{(R_{i-1} - L_{i-1} + 1)\epsilon}{16}.$$

Thus,

$$\sum_{j \in S} \mathbb{I}(h'(X_j) \neq Y_j) \leq \sum_{j \in S} \mathbb{I}(h^*(X_j) \neq Y_j) + \frac{(R_{i-1} - L_{i-1} + 1)\epsilon}{16}.$$

Dividing both sides by $n$, and using the definition of $\text{err}_S$, we obtain:

$$\text{err}_S(h') \leq \text{err}_S(h^*) + \frac{(R_{i-1} - L_{i-1} + 1)}{n} \frac{\epsilon}{16}. \tag{4}$$

From the assumption that the error of $h^*$ on the interval $[L_{i-1}, R_{i-1}]$ is greater than $\frac{1}{16}$, we have:

$$\frac{1}{16} \frac{R_{i-1} - L_{i-1} + 1}{n} \leq \text{err}_S(h^*) \Rightarrow \frac{(R_{i-1} - L_{i-1} + 1)}{n} \frac{\epsilon}{16} \leq \text{err}_S(h^*)\epsilon.$$

Using this inequality and substituting into Inequality 4, we find:

$$\text{err}_S(h') \leq \text{err}_S(h^*)(1 + \epsilon).$$

Hence, the algorithm returns a classifier $h'$ that satisfies the desired error bound. $\qquad\square$

## B.1 Proving Algorithm 1 is correct

In this section we prove Algorithm 1 is correct, meaning it returns a $(1 + \epsilon)$-approximate decision stump with probability at least $1 - \delta$.

*Proof of Theorem 4.2.* We start by noting that, by Lemma B.2, with probability at least $1 - \delta$, all lower/upper bounds calculated during the execution of Algorithm 1 are valid. Furthermore, according to Lemma B.4, if these bounds are valid throughout the execution, the optimal classifier will not be eliminated at any point.

Thus, if the algorithm never enters the *If statement*, it will return the optimal classifier.

On the other hand, if the algorithm does enter the *If statement*, we can reason as follows: From Lemma B.6, we know that entering the *If statement* implies that the error of the optimal classifier in the interval $[L_{i-1}, R_{i-1}]$ is greater than $\frac{1}{16}$.

Furthermore, by Lemma B.7, we know that if the error of the optimal classifier in $X_{[L_i, R_i]}$ exceeds $\frac{1}{16}$, then:

$$\text{err}_S(h') \leq \text{err}_S(h^*) \cdot (1 + \epsilon),$$

where $h^*$ is the optimal classifier and $h'$ is the returned classifier. This ensures that the classifier returned by the algorithm is an acceptable approximation to the optimal classifier.

Thus, we have shown that the algorithm will always return an acceptable classifier, either by directly outputting the optimal classifier or by returning a classifier with an error bounded by $(1 + \epsilon)$ times the error of the optimal classifier.

$\square$

### B.2 ALGORITHM 1 LABEL COMPLEXITY

In this section we prove Theorem 4.1 which bounds the label complexity of Algorithm 1.

*Proof of Theorem 4.1.* By Lemma B.1, we know that the loop in Algorithm 1 will execute at most $\log_2 2n$ times. In each iteration, the algorithm queries at most the following number of labels:

$$c_1 \ln \frac{1}{\delta'} + b_1 = c_1 \ln \left( \frac{2 \log_2 2n}{\delta} \right) + b_1.$$

If the algorithm enters the *If statement*, it will check additional labels of size:

$$\frac{c_2}{\epsilon^2} \left( \ln \frac{1}{\epsilon \delta} + b_2 \right).$$

Thus, the total number of label checks performed is bounded by the sum of the iterations:

$$\log_2 2n \cdot \left( c_1 \ln \left( \frac{2 \log_2 2n}{\delta} \right) + b_1 \right) + \frac{c_2}{\epsilon^2} \left( \ln \frac{1}{\epsilon \delta} + b_2 \right).$$

This expression simplifies to:

$$O \left( \ln n \left( \ln \ln n + \ln \frac{1}{\delta} \right) + \frac{\ln \frac{1}{\epsilon \delta}}{\epsilon^2} \right).$$

$\square$

### B.3 LOWER BOUND ON LABEL COMPLEXITY FOR ACTIVE LEARNING WITH STUMPS

In this subsection, we establish the tightness of the provided algorithm by deriving a lower bound on the number of queries required for active learning with decision stumps, while ignoring logarithmic factors. Specifically, we present Theorem 4.3, which states that at least $O \left( \frac{\ln \frac{1}{\delta}}{\epsilon^2} \right)$ queries are necessary to obtain a $(1 + \epsilon)$-approximate decision stump with a probability greater than $1 - \delta$.

*Proof of Theorem 4.3.* We proceed by applying a well-known result from statistics, which states the following theoremKääriäinen (2006).

**Theorem B.8.** *Given a biased coin with a head probability of either $\frac{1}{2} - \lambda$ or $\frac{1}{2} + \lambda$, at least $\Omega \left( \frac{\ln \frac{1}{\delta}}{\lambda^2} \right)$ coin tosses are required to determine with probability at least $1 - \delta$ which side the coin is biased toward.*

To prove Theorem 4.3, we will leverage Theorem B.8 and demonstrate that the problem of determining the bias of the coin can be reduces to active learning algorithm attempting to solve this problem.

We proceed by contradiction. Assume there exists an algorithm $\mathcal{A}$ that returns a $(1+\epsilon)$-approximate classifier, where the classifier's error rate is less than $\text{err}_S(h^*)(1+\epsilon)$ with probability at least $1 - \delta'$, using fewer than $\ln(\frac{1}{\delta'}) \cdot \frac{1}{\epsilon^2}$ queries.

Now, consider a biased coin whose head probability is either $\frac{1}{2} - \lambda$ or $\frac{1}{2} + \lambda$. We construct the dataset $D = \left\{ \frac{i}{n} \mid 1 \leq i \leq n \right\}$ and assign labels to it as following. For each data point, we toss the coin and report a label of 0 if the coin lands heads, and 1 if it lands tails.

**Claim B.9.** *If $h^*$ is the optimal stump classifier over D, for sufficiently large n we have*

$$P\left(err_S(h^*) \leq \frac{1}{2} - \frac{\lambda}{2}\right) \geq 1 - \frac{\delta}{3}$$

*Proof of Claim B.9.* Without loss of generality, suppose the coin is biased toward heads, meaning the labels are biased toward 0. Let $h_0$ be the classifier that assigns 0 to all points (i.e., its threshold is 1). Then:

$$\mathrm{err}_S(h^*) \leq \mathrm{err}_S(h_0) \quad \Rightarrow \quad P\left(\mathrm{err}_S(h^*) \leq \frac{1}{2} - \frac{\lambda}{2}\right) \geq P\left(\mathrm{err}_S(h_0) \leq \frac{1}{2} - \frac{\lambda}{2}\right).$$

Since $h_0$ misclassify all samples with label 1 and correctly classifies all samples with label 0, we have $\mathrm{err}_S(h_0) = \frac{1}{n} \times$ number of labels 1.

The number of labels equal to 1 follows a Binomial distribution with parameters $n$ and $\frac{1}{2} - \lambda$:

$$\text{number of labels } 1 \sim \mathrm{Binomial}(n, \frac{1}{2} - \lambda).$$

Therefore,

$$P\left(\mathrm{err}_S(h^*) \leq \frac{1}{2} - \frac{\lambda}{2}\right) \geq P\left(\mathrm{err}_S(h_0) \leq \frac{1}{2} - \frac{\lambda}{2}\right) = P\left(\frac{1}{n} \times \mathrm{Binomial}(n, \frac{1}{2} - \lambda) < \frac{1}{2} - \frac{\lambda}{2}\right).$$

We now bound the right-hand side using Chernoff's inequality Bertsekas & Tsitsiklis (2008), which implies:

$$P\left(\mathrm{Binomial}(n, \frac{1}{2} - \lambda) < n\left(\frac{1}{2} - \lambda\right)\left(1 + \frac{\lambda}{1 - 2\lambda}\right)\right) \geq 1 - \exp\left(-\frac{\left(\frac{\lambda}{1-2\lambda}\right)^2 n\left(\frac{1}{2} - \lambda\right)}{2 + \frac{\lambda}{1-2\lambda}}\right).$$

By increasing $n$, we can make this probability greater than $1 - \frac{\delta}{3}$. $\qquad\square$

Now we apply algorithm $\mathcal{A}$ with parameters $\epsilon = \frac{\lambda}{3}$ and $\delta' = \frac{\delta}{3}$ to dataset $D$ to classify based on its labels.

**Claim B.10.** *The algorithm $\mathcal{A}$ will return a classifier with error less than $\left(\frac{1}{2} - \frac{\lambda}{2}\right)(1 + \epsilon)$ with probability at least $1 - \frac{2\delta}{3}$.*

*Proof of Claim B.10.* This follows from the B.9 that with probability at least $1 - \frac{\delta}{3}$, we have $\mathrm{err}_S(h^*) \leq \frac{1}{2} - \frac{\lambda}{2}$, and the fact that algorithm $\mathcal{A}$ returns a classifier with error less than $\mathrm{err}_S(h^*)(1 + \epsilon)$ with probability at least $1 - \frac{\delta}{3}$. As a result, the error of the returned classifier is less than $\left(\frac{1}{2} - \frac{\lambda}{2}\right)(1 + \epsilon)$ with probability at least $\left(1 - \frac{\delta}{3}\right) \cdot \left(1 - \frac{\delta}{3}\right) \geq 1 - \frac{2\delta}{3}$. $\qquad\square$

**Claim B.11.** *If the coin is biased toward heads (i.e., labels are biased toward 0), then for sufficiently large n, all classifiers h with threshold less than $\frac{1}{2}$ have error higher than*

$$\left(\frac{1}{2} - \frac{\lambda}{2}\right)(1 + \epsilon) \quad \text{with probability at least} \quad 1 - \frac{\delta}{3}.$$

*Note that the similar statement holds for the case where the coin is biased toward tails as well.*

*Proof of Claim B.11.* To prove this, we proceed as follows:

$$P\left(\exists_{h \leq \frac{1}{2}} : \mathrm{err}_S(h) \leq \left(\frac{1}{2} - \frac{\lambda}{2}\right)(1 + \epsilon)\right) \leq \sum_{h < \frac{1}{2}} P\left(\mathrm{err}_S(h) \leq \left(\frac{1}{2} - \frac{\lambda}{2}\right)(1 + \epsilon)\right)$$

Substituting $\lambda = 3\epsilon$, we get:

$$\left(\frac{1}{2} - \frac{\lambda}{2}\right)(1 + \epsilon) = \frac{1}{2} - \epsilon - \frac{3}{2}\epsilon^2 \leq \frac{1}{2} - \epsilon.$$

Thus,

$$\sum_{h < \frac{1}{2}} P\left(\text{err}_S(h) \leq \left(\frac{1}{2} - \frac{\lambda}{2}\right)(1 + \epsilon)\right) \leq \sum_{h < \frac{1}{2}} P\left(\text{err}_S(h) \leq \frac{1}{2} - \epsilon\right)$$

For every $h = \frac{i}{n}$ where $i < \frac{n}{2}$, its probability term in the summation can be upper bounded as follows. Let us define: $Z := n(\text{err}_S(h)) = l_i + r_i$, where $l_i \sim \text{Bin}(i, \frac{1}{2} - \lambda)$, $r_i \sim \text{Bin}(n - i, \frac{1}{2} + \lambda)$. Then we have:

$$P\left(\text{err}_S(h) \leq \frac{1}{2} - \epsilon\right) = P\left(Z \leq n(\frac{1}{2} - \epsilon)\right)$$

Using the multiplicative Chernoff lower bound Bertsekas & Tsitsiklis (2008) on the variable $Z$ for $\alpha = 1 - \frac{n(\frac{1}{2} - \epsilon)}{\mu_Z}$ where $\mu_Z = i\left(\frac{1}{2} - \lambda\right) + (n - i)\left(\frac{1}{2} + \lambda\right) = \frac{n}{2} + \lambda(n - 2i)$, this probability is bounded as follows:

$$\Pr[Z \leq (1 - \alpha)\mu_Z] \leq \exp\left(-\frac{1}{2}\mu_Z \alpha^2\right),$$

Substituting $\alpha$ and $\mu_Z$ will result in:

$$\Pr\left[Z \leq (1 - \alpha)\mu_Z\right] = \Pr\left[Z \leq n(\frac{1}{2} - \epsilon)\right] \leq \exp\left(-\frac{1}{2}\mu_Z \alpha^2\right) = \exp\left(-\frac{\left(\mu_Z - n(\frac{1}{2} - \epsilon)\right)^2}{2\mu_Z}\right)$$

Using $\mu_z = \frac{n}{2} + \lambda(n - 2i)$:

$$\exp\left(-\frac{\left(\mu_Z - n(\frac{1}{2} - \epsilon)\right)^2}{2\mu_Z}\right) = \exp\left(-\frac{n\left[\lambda(1 - \frac{2i}{n}) + \epsilon\right]^2}{2[\frac{1}{2} + \lambda(1 - \frac{2i}{n})]}\right) \leq \exp\left(-\frac{n \cdot \epsilon^2}{2(\frac{1}{2} + \lambda)}\right)$$

To summarize the result so far, we have proved that for every $h \leq \frac{1}{2}$:

$$P\left(\text{err}_S(h) \leq \frac{1}{2} - \epsilon\right) \leq \exp\left(-\frac{n \cdot \epsilon^2}{2(\frac{1}{2} + \lambda)}\right)$$

Finally, doing a summation over all $h$ will get to:

$$\sum_{h < \frac{1}{2}} P\left(\text{err}_S(h) \leq \frac{1}{2} - \epsilon\right) \leq \frac{n}{2}\exp\left(-\frac{n \cdot \epsilon^2}{2(\frac{1}{2} + \lambda)}\right) \leq \frac{\delta}{3}$$

The last inequality holds for any sufficiently large $n$, because $\frac{n}{2}$ grows linearly but $\exp\left(-\frac{n \cdot \epsilon^2}{2(\frac{1}{2} + \lambda)}\right)$ decreases exponentially, making the entire term as small as desired. $\qquad\square$

Now, based on Claim B.10 the algorithm $\mathcal{A}$ returns a classifier with an error rate of less than

$$\left(\frac{1}{2} - \frac{\lambda}{2}\right)(1 + \epsilon)$$

with a probability greater than $1 - \frac{2}{3}\delta$. Moreover, based on Claim B.11 all classifiers on the wrong side of $\frac{1}{2}$ have an error greater than $(\frac{1}{2} - \frac{\lambda}{2})(1 + \epsilon)$ with probability greater than $1 - \frac{\delta}{3}$. Thus, with

the probability at least $(1 - \frac{2}{3}\delta)(1 - \frac{\delta}{3}) \geq 1 - \delta$, the returned hypothesis $h$ can indicate whether the coin is biased toward heads or tails, by choosing "heads" if $h > \frac{1}{2}$ and "tails" if $h \leq \frac{1}{2}$. However, as established in Theorem B.8, any algorithm requires at least $\Omega(\frac{\ln\left(\frac{1}{\delta}\right)}{\lambda^2})$ samples. Therefore, the algorithm $\mathcal{A}$ needs at least:

$$\Omega(\frac{\ln\left(\frac{3}{\delta}\right)}{\epsilon^2})$$

samples to achieve this. $\square$

## C  GENERAL BINARY CLASSIFICATION PROOFS

In this section, we provide the proofs corresponding to Section 4.3, where we extend our algorithm to general binary classification tasks as described in Algorithm 2. The structure of the proofs closely follows the approach in Appendix B.

To facilitate understanding the general classification algorithm, we first presented the case for stumps. Table 2 outlines the correspondence between lemmas and theorems in the stump case and their general binary classification counterparts.

Table 2: Correspondence between Stump and General Binary Classification results.

| Description | Stump Version | General Binary Classification Version |
|---|---|---|
| Body section | Section 4.1 | Section 4.3 |
| Proofs in Appendix | Appendix B | Appendix C |
| Algorithm | Algorithm 1 | Algorithm 2 |
| Algorithm Correctness | Theorem 4.2 | Theorem 4.4 |
| Time Complexity | Theorem 4.1 | Theorem 1.2 |
| Number of Iterations | Lemma B.1 | Lemma C.1 |
| Bounds Validity | Lemma B.2 | Lemma C.2 |
| Error Comparison | Lemma B.3 | Lemma B.3 |
| Optimal Classifier Not Eliminated | Lemma B.4 | Lemma C.5 |
| Lower Bound Error with $h^*$ | Lemma B.5 | Lemma C.6 |
| Low Optimal Error $\rightarrow$ Reiterate | Lemma B.6 | Lemma C.7 |
| High Optimal Error $\rightarrow$ Correct Output | Lemma B.7 | Lemma C.8 |

We begin with the following lemma, which establishes that the maximum number of iterations in the loop of Algorithm 2 is bounded by $\log_2(2n)$.

**Lemma C.1.** *Algorithm 2 will execute the loop at most $\log_2 2n$ times.*

*Proof of Lemma C.1.* Initially, we have $r_0 = 1$. During each iteration, if the inequality $r_i > \frac{r_{i-1}}{2}$ holds, the algorithm terminates immediately. Thus, for the loop to continue, it must be that $r_i \leq \frac{r_{i-1}}{2}$.

If at any point $r_i < \frac{1}{n}$, no two classifiers in the set $H_i$ can disagree on any samples, leaving only a single classifier to be considered. In this scenario, the algorithm will again conclude. Therefore, the execution of the loop cannot surpass the threshold of iterations where $r_i$ becomes smaller than $\frac{1}{n}$.

Consequently, the number of iterations required is at most:

$$1 + \log_2 n = \log_2 2n$$

$\square$

We need to show that all lower and upper bounds are valid during the algorithm's execution with probability at least $1 - \delta$, simultaneously.

**Lemma C.2.** *In an execution of Algorithm 2, all estimated lower and upper bounds are valid with probability at least $1 - \delta$*

*Proof of Lemma C.2.* From Lemma C.1, we know that the bounds outside the *If statement* are evaluated at most $\log_2(2n)$ times. Each time, the bounds are correct with probability at least $1 - \delta'$. Thus, the probability of a bound being incorrect is less than $\delta'$. Therefore, the probability of at least one bound being incorrect is less than $\delta' \log_2(2n)$. Using the definition of $\delta'$, we know $\delta' = \frac{\delta}{2\log_2(2n)}$, so we have $\delta' \log_2(2n) = \frac{\delta}{2}$.

Additionally, the bounds inside the *If statement* are evaluated only once, and the probability of them being wrong is less than $\frac{1}{2\delta}$. Combining these two factors, the overall probability of any bound being incorrect is less than $\delta$. $\square$

The following lemma establishes that if $h^* \in H'$, then $H' \subseteq B_H(h^*, 2\text{radius}(H'))$. Consequently, this implies that during iteration $i$, when the radius is $\text{radius}(H_i)$, all classifiers in $H_i$ are at most a distance of $2\text{radius}(H_i)$ from $h^*$.

**Lemma C.3.** *If $H' \subseteq B_H(h, r)$ and $h' \in H'$, then $H' \subseteq B_H(h', 2r)$.*

*Proof of Lemma C.3.* We aim to show that $B_H(h, r) \subseteq B_H(h', 2r)$. Consider any $h'' \in B_H(h, r)$. By definition, we have: $D_S(h, h'') \leq r$ which implies: $r \geq \frac{1}{n}\sum_{x \in S} \mathbb{I}(h(x) \neq h''(x))$. Similarly, since $h' \in B_H(h, r)$, we have: $D_S(h, h') \leq r$, which implies: $r \geq \frac{1}{n}\sum_{x \in S} \mathbb{I}(h(x) \neq h'(x))$. Adding these inequalities gives:

$$2r \geq \frac{1}{n}\sum_{x \in S} \left(\mathbb{I}(h(x) \neq h'(x)) + \mathbb{I}(h(x) \neq h''(x))\right).$$

Notice that: $\mathbb{I}(h(x) \neq h'(x)) + \mathbb{I}(h(x) \neq h''(x)) \geq \mathbb{I}(h'(x) \neq h''(x))$ Therefore, we have: $2r \geq \frac{1}{n}\sum_{x \in S} \mathbb{I}(h'(x) \neq h''(x))$. Thus, by definition of $D_S$, we conclude: $2r \geq D_S(h', h'')$

Hence, any $h'' \in B_H(h, r)$ is also in $B_H(h', 2r)$. $\square$

The following lemma helps us relate $D_S(h, h')$ to $D_{\text{DIS}(H')}(h, h')$. This relation is important because the final error is measured in $S$, but we randomly sample from $\text{DIS}(H')$, which leads to bounds on $D_{\text{DIS}(H')}$.

**Lemma C.4.** *If $D_S(h, h') \geq \frac{r}{2}$ for some $r$, and $h, h' \in H'$ where $H' \subseteq B_H(h, 2r)$, then*

$$D_{DIS(H')}(h, h') \geq \frac{nr}{|B_H(h, 2r)|} \cdot \frac{1}{2}.$$

*Proof of Lemma C.4.* Given that all samples where $h$ and $h'$ disagree are in $\text{DIS}(H')$, the number of disagreements in $\text{DIS}(H')$ is equal to those in $S$. From the definition of $D_S$ and since all disagreements are included, we know:

$$D_{\text{DIS}(H')}(h, h') = \frac{1}{|\text{DIS}(H')|}\sum_{x \in \text{DIS}(H')} \mathbb{I}(h(x) \neq h'(x)) = \frac{1}{|\text{DIS}(H')|}\sum_{x \in S} \mathbb{I}(h(x) \neq h'(x))$$

Thus,

$$D_{\text{DIS}(H')}(h, h') = \frac{|S|}{|\text{DIS}(H')|}\left(\frac{1}{|S|}\sum_{x \in S}\mathbb{I}(h(x) \neq h'(x))\right) = \frac{|S|}{|\text{DIS}(H')|}D_S(h, h')$$

By assumption $D_S(h, h') \geq \frac{r}{2}$ and since $H' \subseteq B_H(h, 2r)$, it follows:

$$D_{\text{DIS}(H')}(h, h') \geq \frac{nr}{|\text{DIS}(B_H(h, 2r))|} \cdot \frac{1}{2}$$

$\square$

The following lemma ensures that no optimal classifiers are eliminated if all bounds during the algorithm's execution are correct.

**Lemma C.5.** *If all bounds during the execution of Algorithm 2 are valid, the algorithm will not eliminate any optimal classifiers if all lower/upper bounds are valid.*

*Proof of Lemma C.5.* We use a proof by contradiction. Suppose there is an iteration $i$ where an optimal classifier $h^*$ is eliminated despite all bounds being valid. This implies $h^* \in H_i$ but $h^* \notin H_{i+1}$, which means: $\text{LB}(S_i, h^*, \delta') > \min_{h \in H_i} \text{UB}(S_i, h, \delta')$. Suppose $h'$ achieves the minimum: $\min_{h \in H_i} \text{UB}(S_i, h, \delta') = \text{UB}(S_i, h', \delta')$. Thus, we have: $\text{LB}(S_i, h^*, \delta') > \text{UB}(S_i, h', \delta')$

Let $S' = \text{DIS}(H_i)$. With the validity of bounds:

$$\text{err}_{S'}(h^*) \geq \text{LB}(S_i, h^*, \delta') > \text{UB}(S_i, h', \delta') \geq \text{err}_{S'}(h') \tag{5}$$

Since $h^*$ is optimal, we know $\text{err}_S(h^*) \leq \text{err}_S(h')$. From Lemma B.3, this implies: $\text{err}_{S'}(h^*) \leq \text{err}_{S'}(h')$. This contradicts inequality 5, thus proving that an optimal classifier cannot be eliminated if all bounds are valid. $\qed$

The following lemma bounds the error of a classifier $h$ based on its distance from the optimal classifier $h^*$ and the error of $h^*$. Specifically, it shows that if $h$ is far from $h^*$ and $h^*$ has low error, the error of $h$ must be high, leading $h$ to be eliminated.

**Lemma C.6.** *For all $h, h^* \in H'$, the following inequality holds:*

$$err_{DIS(H')}(h) \geq D_{DIS(H')}(h, h^*) - err_{DIS(H')}(h^*).$$

*Proof of Lemma C.6.* From definition of the error function $\text{err}_{\text{DIS}(H')}(h)$:

$$\text{err}_{\text{DIS}(H')}(h) = \frac{1}{|\text{DIS}(H')|} \sum_{j \in \text{DIS}(H')} \mathbb{I}(h(X_j) \neq Y_j),$$

Define $S'$ as the set of samples $h$ and $h^*$ makes different predictions. So $S' = \{x \in S \mid h(x) \neq h^*(x)\}$. Since all samples that $h$ and $h^*$ makes different predictions are in $\text{DIS}(H')$, $S' \subseteq \text{DIS}(H')$. Assume $h$ makes $M$ misclassifications in $\text{DIS}(H')/S'$. Since $h^*$ behave identical to $h$ on these samples, $h^*$ also make $M$ misclassifications in $\text{DIS}(H')/S'$.

Thus, the error of $h$ can be expressed as:

$$\text{err}_{\text{DIS}(H')}(h) = \frac{1}{|\text{DIS}(H')|} \left( \sum_{j \in S'} \mathbb{I}(h(X_j) \neq Y_j) + M \right).$$

Because $h$ and $h^*$ make different predictions on $X_j$ for $j \in S'$, we have:

$$\text{err}_{\text{DIS}(H')}(h) = \frac{1}{|\text{DIS}(H')|} \left( \left( \sum_{j \in S'} 1 - \mathbb{I}(h^*(X_j) \neq Y_j) \right) + M \right).$$

Here, $\sum_{j \in S'} 1$ counts the total number of samples in $S'$, while $\sum_{j \in S'} \mathbb{I}(h^*(X_j) \neq Y_j)$ counts the number of misclassifications made by $h^*$ in $S'$. Simplifying further:

$$\text{err}_{\text{DIS}(H')}(h) = \frac{|S'|}{|\text{DIS}(H')|} - \frac{1}{|\text{DIS}(H')|} \left( \sum_{j \in S'} \mathbb{I}(h^*(X_j) \neq Y_j) + M \right) + \frac{2M}{|\text{DIS}(H')|}. \tag{6}$$

Given that $M$ represents the number of misclassifications made by $h^*$ outside $S'$, the error of $h^*$ can be expressed as:

$$\text{err}_{\text{DIS}(H')}(h^*) = \frac{1}{|\text{DIS}(H')|} \left( \sum_{j \in S'} \mathbb{I}(h^*(X_j) \neq Y_j) + M \right).$$

Substituting this into Equality 6:

$$\text{err}_{\text{DIS}(H')}(h) = \frac{|S'|}{|\text{DIS}(H')|} - \text{err}_{\text{DIS}(H')}(h^*) + \frac{2M}{|\text{DIS}(H')|}.$$

Since $M > 0$, it follows that:

$$\text{err}_{\text{DIS}(H')}(h) \geq \frac{|S'|}{|\text{DIS}(H')|} - \text{err}_{\text{DIS}(H')}(h^*).$$

This completes the proof of Lemma C.6. □

Having the above Lemmas in place we provide the two main following lemmas. The following Lemma C.7 shows if the error of optimal classifier is low in $\text{DIS}(H_i)$ the algorithm will reiterate the for loop.

**Lemma C.7.** *There exist universal constants $c_1, b_1$ such that for any iteration of Algorithm 2, if*

$$err_{DIS(H_i)}(h^*) \leq \frac{1}{16\theta}$$

*then the radius$(H_{i+1}) \leq \frac{1}{2}$radius$(H_i)$, provided that all lower and upper bounds are valid during the algorithm's execution.*

*Proof of Lemma C.7.* $r_i$ is defined as radius$(H_i)$. Then using Lemma C.4 we know for all $h \in H_i$ that $D_S(h, h^*) \geq \frac{r_i}{2}$ we have $D_{\text{DIS}(H_i)}(h^*, h) \geq \frac{nr_i}{|B_H(h^*, 2r_i)|} \cdot \frac{1}{2}$. Using Lemma C.6 we know that

$$\text{err}_{\text{DIS}(H')}(h) \geq D_{\text{DIS}(H')}(h, h^*) - \text{err}_{\text{DIS}(H')}(h^*),$$

plunging $D_{\text{DIS}(H')}(h, h^*) \geq \frac{nr_i}{|B_H(h^*, 2r_i)|} \cdot \frac{1}{2}$ and $\text{err}_{\text{DIS}(H')}(h^*) \leq \frac{nr}{|B_H(h^*, 2r_i)|} \cdot \frac{1}{16}$ we get,

$$\text{err}_{\text{DIS}(H')}(h) \geq \frac{nr_i}{|B_H(h^*, 2r_i)|} \cdot \frac{1}{2} - \frac{nr_i}{|B_H(h^*, 2r_i)|} \cdot \frac{1}{16} = \frac{nr_i}{|B_H(h^*, 2r_i)|} \cdot \frac{7}{16}.$$

Using definition of $\theta$ in Definition 3.3 we get

$$\text{err}_{\text{DIS}(H')}(h) \geq \frac{7}{16\theta}.$$

From Theorem A.1 we know that if we have

$$|S'| = \frac{64}{(\frac{1}{16\theta})^2} \left( 2V_H \ln(12 \cdot 16\theta) + \ln(\frac{4}{\delta'}) \right) \in O\left( \theta^2 (V_H \ln(\theta) + \ln(\frac{1}{\delta'})) \right),$$

then $\text{err}_{\text{DIS}(H')}(h) - \text{LB}(S', h, \delta') \leq \frac{1}{16\theta}$ Therefore $\text{LB}(S', h, \delta') \geq \frac{6}{16\theta}$

Similarly since we assumed $\text{err}_{S'}(h^*) \leq \frac{1}{16\theta}$, and we have $\text{UB}(S', h^*, \delta') \leq \text{err}_{S'}(h^*) + \frac{1}{16\theta}$ we have $\text{UB}(S', h^*, \delta') \leq \frac{2}{16}$, therefore $\text{UB}(S', h^*, \delta') < \text{LB}(S', h, \delta')$.

So there exist $c_1, b_1$ that all classifiers like $h$ that $D_S(h, h^*) \geq \frac{r_i}{2}$ will be removed from the $H_i$ thus, radius$(H_{i+1}) \leq \frac{r_i}{2}$. □

**Lemma C.8.** *There exist universal constants $c_2$ and $b_2$ such that for any iteration of Algorithm 2, if*

$$err_{DIS(H_i)}(h^*) > \frac{1}{16\theta},$$

*and the algorithm enters the If statement, it will return a classifier like $h'$ where*

$$err_S(h') \leq err_S(h^*)(1 + \epsilon),$$

*provided that all lower and upper bounds are valid during the algorithm's execution.*

*Proof of Lemma C.8.* The Algorithm 2 will build a set $S'$ consists of

$$\frac{c_2\theta^2}{\epsilon^2}\left(V_H \ln\left(\frac{\theta}{\epsilon}\right) + \ln\frac{1}{\delta}\right) + b_2$$

random samples drawn from $\text{DIS}(H_i)$.

From Theorem A.1 we know using $\frac{64}{(\frac{\epsilon}{16\theta})^2}\left(2V_H \ln(\frac{12}{\frac{\epsilon}{16\theta}}) + \ln(\frac{4}{\frac{\delta}{2}})\right)$ samples we get bounds such that $\text{UB}(S', h, \frac{\delta}{2}) \leq \text{err}_{\text{DIS}(H_i)}(h) + \frac{\epsilon}{16\theta}$ for all $h$. Therefore, there exists universal $c_2$ and $b_2$ such that this bound holds.

The Algorithm returns $h' = \arg\min_{h \in H_i} \text{UB}(S', h, \frac{\delta}{2})$. Therefore, we have $\text{err}_{\text{DIS}(H_i)}(h') \leq \text{err}_{\text{DIS}(H_i)}(h^*) + \frac{\epsilon}{16\theta}$. Given that $\text{err}_{\text{DIS}(H_i)}(h^*) \geq \frac{1}{16\theta}$, we have

$$\text{err}_{\text{DIS}(H_i)}(h') \leq \text{err}_{\text{DIS}(H_i)}(h^*)(1 + \epsilon). \tag{7}$$

Since $h' \in H_i$ and $h^* \in H_i$ $\text{DIS}(H_i)$ include all samples they label differently. Assume they misclassify $M$ samples in $S/\text{DIS}(H_i)$.

$$\text{err}_S(h^*) = \frac{1}{n} \sum_{(x,y) \in S} \mathbb{I}(h^*(x) \neq y)$$

$$= \frac{1}{n}\left(\sum_{(x,y) \in \text{DIS}(H_i)/S} \mathbb{I}(h^*(x) \neq y) + \sum_{(x,y) \in \text{DIS}(H_i)} \mathbb{I}(h^*(x) \neq y)\right)$$

$$= \frac{1}{n}(M + |\text{DIS}(H_i)|\text{err}_{\text{DIS}(H_i)}(h^*))$$

Multiplying both side by $1 + \epsilon$ we get,

$$(1 + \epsilon)\text{err}_S(h^*) = \frac{1}{n}\left((1 + \epsilon)M + (1 + \epsilon)|\text{DIS}(H_i)|\text{err}_{\text{DIS}(H_i)}(h^*)\right)$$

Since $M \geq 0$

$$(1 + \epsilon)\text{err}_S(h^*) \geq \frac{1}{n}(M + |\text{DIS}(H_{i-1})|(1 + \epsilon)\text{err}_{\text{DIS}(H_{i-1})}(h^*)) \tag{8}$$

Applying Inequality 7 to Inequality 8 we have

$$(1 + \epsilon)\text{err}_S(h^*) \geq \frac{1}{n}(M + |\text{DIS}(H_{i-1})|\text{err}_{\text{DIS}(H_{i-1})}(h')) = \text{err}_S(h')$$

$\square$

Now lets proof Algorithm 2 correctness.

*Proof of Theorem 4.4.* First, in Lemma C.1, we establish that the loop in Algorithm 2 runs for at most $\log_2(2n)$ iterations. Using this result, we show in Lemma C.2 that all bounds are satisfied with probability at least $1 - \delta$, ensuring that we can safely assume all lower and upper bounds are valid during the algorithm's execution. Next, we prove that the optimal classifier, $h^*$, is never removed, as shown in Lemma C.5, assuming that all bounds hold. Then in Lemma C.7 we show that if $\text{err}_{\text{DIS}(H_i)}(h^*) \leq \frac{1}{16\theta}$ then we will no go into the *If statement*. Finally in Lemma C.8 we show that if $\text{err}_{\text{DIS}(H_i)}(h^*) > \frac{1}{16\theta}$ and we do go into the *If statement* then the algorithm will return a $(1 + \epsilon)$ classifier. $\square$

*Proof of Theorem 1.2.* Theorem 4.4 let us show that Algorithm 2 returns a $(1 + \epsilon)$-approximate classifier with probability greater than $1 - \delta$. For its label complexity, we apply Lemma C.1 to show that the loop in the algorithm repeats at most $\log_2 2n$ times, and since the *If statement* is executed only once, the label complexity is bounded by $\ln(n)$ times $O(\theta^2(V_H \ln\theta + \ln\frac{1}{\delta'}))$ plus $O\left(\frac{\theta^2}{\epsilon^2}(V_H \ln\frac{\theta}{\epsilon} + \ln\frac{1}{\delta})\right)$. This concludes the proof of Theorem 1.2. $\square$

# D  DECISION TREE'S $\theta$ CALCULATION

As the first lemma, we prove that DIS can be expressed as a relationship between a single classifier and the other classifiers in the disagreement set.

**Lemma D.1.** *Assuming $h \in H$, we have*

$$DIS(H) = \{x \mid \exists_{h' \in H} : h'(x) \neq h(x)\}.$$

*Proof of Lemma D.1.* From the definition, we have:

$$\text{DIS}(H) = \{x \mid \exists h_1, h_2 \in H : h_1(x) \neq h_2(x)\}$$

If for some $x$, we have $h_1(x) \neq h_2(x)$, then either $h(x) \neq h_1(x)$ or $h(x) \neq h_2(x)$. Therefore, if $\exists h_1, h_2 \in H : h_1(x) \neq h_2(x)$, then $\exists h' \in H : h'(x) \neq h(x)$.  □

Next, in the following Lemma, we build a connection between decision trees and line trees.

**Lemma D.2.** *For any two decision trees $h, h'$ and any two leaves $i, j$ such that $l_{h,i} \neq l_{h',j}$, if $S_i = \{x \mid LineTree_{h,i}(x) = l_{h,i}\}$, then:*

$$D_{S_i}(LineTree_{h,i}, LineTree_{h',j}) \leq D_{S_i}(h, h')$$

*Proof of Lemma D.2.* If for some $x$ we have $\text{LineTree}_{h',j}(x) = l_{h',j}$, then $h'(x) = l_{h',j}$. Therefore:

$$\{x \in S_i \mid \text{LineTree}_{h',j}(x) = l_{h',j}\} \subseteq \{x \in S_i \mid h'(x) = l_{h',j}\}$$

Since $l_{h,i} \neq l_{h',j}$, we have:

$$\{x \in S_i \mid \text{LineTree}_{h',j}(x) \neq l_{h,i}\} \subseteq \{x \in S_i \mid h'(x) \neq l_{h,i}\}$$

Since $x \in S_i$, we have $\text{LineTree}_{h,i}(x) = l_{h,i}$. As a result $h(x) = l_{h,i}$. Therefore:

$$\{x \in S_i \mid \text{LineTree}_{h,i}(x) \neq \text{LineTree}_{h',j}(x)\} \subseteq \{x \in S_i \mid h(x) \neq h'(x)\}$$

Thus:

$$|\{x \in S_i \mid \text{LineTree}_{h,i}(x) \neq \text{LineTree}_{h',j}(x)\}| \leq |\{x \in S_i \mid h(x) \neq h'(x)\}|$$

Thus:

$$|S_i| D_{S_i}(\text{LineTree}_{h,i}, \text{LineTree}_{h',j}) \leq |S_i| D_{S_i}(h, h')$$

Therefore:

$$D_{S_i}(\text{LineTree}_{h,i}, \text{LineTree}_{h',j}) \leq D_{S_i}(h, h')$$

□

In the following Lemma we relate error of a classifier in the overall dataset to the error of the classifier in a subset.

**Lemma D.3.** *If $S' \subset S$, then $B_S(h, r) \subseteq B_{S'}(h, r\frac{|S|}{|S'|})$.*

*Proof of Lemma D.3.* Assume $h' \in B_S(h, r)$. We will show $h' \in B_{S'}(h, r\frac{|S|}{|S'|})$. We have:

$$D_S(h, h') \leq r$$

Therefore:

$$|\{x \in S \mid h(x) \neq h'(x)\}| \leq r|S|$$

Since $S' \subseteq S$:

$$|\{x \in S' \mid h(x) \neq h'(x)\}| \leq |\{x \in S \mid h(x) \neq h'(x)\}| \leq r|S|$$

Therefore:

$$\frac{1}{|S'|}|\{x \in S' \mid h(x) \neq h'(x)\}| \leq r\frac{|S|}{|S'|}$$

From definition of $D$ right side is equal to $D_{S'}(h, h')$, Thus:

$$D_{S'}(h, h') \leq r\frac{|S|}{|S'|}$$

Therefore:

$$h' \in B_{S'}(h, r\frac{|S|}{|S'|})$$

$\square$

To extend our analysis to line trees which we need for Theorem 1.1, we first introduce some key definitions related to line trees.

**Definition D.4.** A line tree is a decision tree where for each node, at least one of its children is a leaf, and all leaves except the deepest leaf assign the same label, while the deepest leaf assigns the opposite label.

**Definition D.5.** If $h$ is a line tree, then $l_h$ is the label that could be the deepest leaf label and is different from the rest of the leaves' labels.

**Definition D.6.** If $h$ is a line tree, then $d_h \subseteq \{1, 2, \ldots, \dim\}$ is the set of dimensions that nodes in the line tree decide based on.

**Definition D.7.** $\mathbb{L}$ is the set of all line trees with depth less than $d$ where each node decides based on a unique dimension. $\mathbb{L}_{d'}$ is the set of all line trees with depth less than $d$ and with $d_h = d'$.

**Definition D.8.** For a line tree $h \in \mathbb{L}_{d'}$, we define a function $f_h : d_h \to \{\text{prefix}, \text{suffix}\}$, where $f_h(a)$ specifies the splitting behavior of $h$ for each dimension $a \in d_h$:

- $f_h(a) = \text{prefix}$ if samples $x$ with $x_a$ less than a threshold are directed to the leaf with label $l_h$.

- $f_h(a) = \text{suffix}$ if samples $x$ with $x_a$ greater than a threshold are directed to the leaf with label $l_h$.

**Definition D.9.** For a line tree $h \in \mathbb{L}$, let $h_S^a$ denote the number of distinct values of $x_a$ for which $h(x) = l_{h'}$. Formally:
$$h_S^a = |\{x_a \mid x \in S \wedge h(x) = l_h\}|.$$

In the following theorem, we prove that the disagreement coefficient of a decision tree is $O(\ln^d(n))$, assuming the input distribution is uniform-like and each node in a root to leaf path works with a unique dimension.

*Proof of Theorem 1.1 upperbound.* Assume we have chosen a tree $h$ and we want to bound $\theta_h$. This requires bounding $\frac{|\text{DIS}(B_H(h,r))|}{nr}$ for all $r$. Using Lemma D.1, we have:

$$\text{DIS}(B_H(h, r)) = \{x \mid \exists h' \in B_H(h, r) : h'(x) \neq h(x)\}$$

Breaking this set based on the leaf $x$ reaches in $h$, we get:

$$\text{DIS}(B_H(h, r)) = \bigcup_{i=1}^{L}\{x \mid x \text{ reaches leaf } i \text{ in } h \wedge \exists h' \in B_H(h, r) : h'(x) \neq h(x)\}$$

Using the definition of a line tree, $\text{DIS}(B_H(h, r))$ is equivalent to:

$$\bigcup_{i=1}^{L}\{x \mid \text{LineTree}_{h,i}(x) = l_{h,i} \wedge \exists h' \in B_H(h, r) : h'(x) \neq h(x)\}$$

Further splitting the set based on the dimension set of the leaf that $x$ reaches in $h'$, $\text{DIS}(B_H(h, r))$ is equal to:

$$\bigcup_{i=1}^{L} \bigcup_{d' \subset \{1,2,\ldots,\dim\}} \{x \mid \text{LineTree}_{h,i}(x) = l_i \wedge \exists h' \in B_H(h, r), j : x \text{ reaches leaf } j \text{ in } h' \wedge h'(x) \neq h(x) \wedge d_{h',j} = d'\}$$

$$= \bigcup_{i=1}^{L} \bigcup_{d' \subset \{1,2,\ldots,\dim\}} \{x \mid \text{LineTree}_{h,i}(x) = l_i \wedge \exists h' \in B_H(h, r), j : \text{LineTree}_{h',j}(x) = l_{h',j} \wedge h'(x) \neq h(x) \wedge d_{h',j} = d'\}$$

Let $S_i$ be the set of data points that reach leaf $i$ in tree $h$. Formally, $S_i = \{x \mid \text{LineTree}_{h,i}(x) = l_{h,i}\}$.

Then we have:

$$\text{DIS}(B_H(h, r)) \subseteq \bigcup_{i=1}^{L} \bigcup_{d' \subset \{1,2,\ldots,\dim\}} \{x \in S_i \mid \exists h' \in B_H(h, r), j : \text{LineTree}_{h',j}(x) = l_{h',j} \wedge h'(x) \neq h(x) \wedge d_{h',j} = d'\}$$

Using Lemma D.3, this is equal to:

$$= \bigcup_{i=1}^{L} \bigcup_{d' \subset \{1,2,\ldots,\dim\}} \{x \in S_i \mid \exists h' \in B_{H,S_i}(h, r\frac{n}{|S_i|}), j : \text{LineTree}_{h',j}(x) = l_{h',j} \wedge h'(x) \neq h(x) \wedge d_{h',j} = d'\}$$

Using the definitions of line trees, rather than first selecting a general decision tree $h'$ and subsequently addressing one of its line trees, we can directly consider $h'$ as a line tree, significantly simplifying the analysis.

Additionally, observe that if for some $h', j$, we have $D_{S_i}(h, h') \leq r\frac{n}{|S_i|}$ and $l_{h,i} \neq l_{h',j}$, then by Lemma D.2, it follows that $D_{S_i}(h, \text{LineTree}_{h',j}) \leq r\frac{n}{|S_i|}$. Consequently, we can further refine our expression as:

$$\text{DIS}(B_H(h, r)) \subseteq \bigcup_{i=1}^{L} \bigcup_{d' \subset \{1,2,\ldots,\dim\}} \{x \in S_i \mid \exists h' \in B_{\mathbb{L}_{d'},S_i}(h, r\frac{n}{|S_i|}) : h'(x) = l_{h'} \wedge h'(x) \neq h(x)\} \subseteq$$

$$\bigcup_{i=1}^{L} \bigcup_{d' \subset \{1,2,\ldots,\dim\}} \{x \in S_i \mid \exists h' \in B_{\mathbb{L}_{d'},S_i}(h, r\frac{n}{|S_i|}) : h'(x) \neq h(x)\} =$$

Since we only focused on $S_i$, and in $S_i$ $h$ and $\text{LineTree}_{h,i}$ behave similarly, we have:

$$\bigcup_{i=1}^{L} \bigcup_{d' \subset \{1,2,\ldots,\dim\}} \{x \in S_i \mid \exists h' \in B_{\mathbb{L}_{d'},S_i}(\text{LineTree}_{h,i}, r\frac{n}{|S_i|}) : h'(x) \neq \text{LineTree}_{h,i}(x)\}$$

Since $h'$ is a line tree and we only have $\text{LineTree}_{h,i}$, and in $S_i$ $\text{LineTree}_{h,i}$ is an all-same classifier, we can apply Proposition 3.5. Hence, the size of each of the inner sets is of $O\left(|S_i| r\frac{n}{|S_i|}(3 \ln w)^d\right)$.

Since $L \leq 2^d$ and $d'$ has $\binom{\dim}{d}$ choices, the total size of $\text{DIS}(B_H(h, r))$ is of:

$$|\text{DIS}(B_H(h, r))| \leq O\left(2^d \binom{\dim}{d} nr(3 \ln w)^d\right) = O\left(2^d \binom{\dim}{d} nr \left(\frac{3}{\dim} \ln n\right)^d\right)$$

$$\leq O\left(nr6^d \frac{\dim^d}{d!} \left(\frac{1}{\dim} \ln n\right)^d\right) = O\left(nr\frac{6^d}{d!} \ln^d n\right) \leq O\left(nr \ln^d n\right)$$

Therefore:

$$\theta_h(r) \in O\left(\ln(n)^d\right)$$

$\square$

We need the following Lemmas to prove Proposition 3.5.

**Lemma D.10.** *Let $h \in \mathbb{L}_{d'}$ be a line tree. Then:*

$$|\{x \in S \mid h(x) = l_h\}| = \prod_{a \in d'} h_S^a \cdot \prod_{a \notin d'} w_a,$$

*where*

$$S = \{(a_1, \ldots, a_{dim}) \mid \forall i, \, a_i \in \mathbb{N}, \, a_i \leq w_i \leq w\}.$$

*Proof of Lemma D.10.* For each dimension $a \in d'$, exactly $h_S^a$ of the possible values of $x_a$ are directed to the leaf with label $l_h$. By the definition of a line tree (Definition D.4 and Definition D.5), if any $x_a$ does not lead to this leaf, the resulting label for $h(x)$ will be $1 - l_h$, since there is only one leaf with the label $l_h$ in a line tree.

Given that $S$ represents all possible points, and each combination of valid values of $x_a$ corresponds to exactly one point in $S$, the number of points where $h(x) = l_h$ is the product of $h_S^a$ across all dimensions $a \in d'$.

Thus:

$$|\{x \in S \mid h(x) = l_h\}| = \prod_{a \in d'} h_S^a \cdot \prod_{a \notin d'} w_a.$$

$\square$

**Lemma D.11.** *Let $h$ be a line tree with $h(x) = l_h$. Then for any $x'$ satisfying:*

$$\begin{cases} x'_a \leq x_a & \text{if } f_h(a) = \text{prefix} \\ x_a \leq x'_a & \text{if } f_h(a) = \text{suffix} \end{cases} \quad \forall a \in d_h,$$

*we have $h(x') = l_h$.*

*Proof of Lemma D.11.* We know that each node in $h$ directs $x$ toward the leaf labeled $l_h$. We will show that each node also directs $x'$ to the same child.

Consider a node working with dimension $a \in d_h$.

- If $f_h(a) = \text{prefix}$, then values lower than $x_a$ will also be directed toward the leaf labeled $l_h$. Since $x'_a \leq x_a$, $x'$ follows the same path as $x$.

- If $f_h(a) = \text{suffix}$, then values greater than $x_a$ will also be directed toward the leaf labeled $l_h$. Since $x_a \leq x'_a$, $x'$ follows the same path as $x$.

Therefore, in all nodes, $x'$ follows the same path as $x$, and hence acquires the same label $l_h$. $\square$

**Lemma D.12.** *The number of sequences of the form $\langle x_1, x_2, \ldots, x_k \rangle$ such that $\forall_{1 \leq i \leq k} : x_i \in \mathbb{N}$, $\forall_{1 \leq i \leq k} : x_i \leq w_i$, and $\prod_{1 \leq i \leq k} x_i \leq s$ is less than $s \prod_{2 \leq i \leq k} \ln(w_i) + 1$.*

*Proof of Lemma D.12.* We will use induction to prove this lemma. Let $g(s, k, w)$ denote the number of such sequences.

**Base Case:** For $k = 1$, the theorem is obvious since there are only $s$ possible sequences.

**Inductive Step:** Assume the theorem holds for any $s$ and $k = k_0$. We need to prove it for $k = k_0 + 1$.

Consider the possible values of $x_{k_0+1}$. We have:

$$g(s, k_0 + 1, w) = \sum_{1 \leq i \leq w_{k_0+1}} g\left(\frac{s}{i}, k_0, w\right)$$

By the induction hypothesis, this sum is less than:

$$\leq \sum_{1 \leq i \leq w_{k_0+1}} \left(\frac{s}{i} \prod_{2 \leq j \leq k_0} \ln(w_j) + 1\right) = s \left(\prod_{2 \leq j \leq k_0} \ln(w_j) + 1\right) \sum_{1 \leq i \leq w_{k_0+1}} \frac{1}{i}$$

Using the harmonic series approximation, $\sum_{1 \leq i \leq w_{k_0+1}} \frac{1}{i} \leq \ln(w_{k_0+1}) + 1$, we have:

$$\leq s \left( \prod_{2 \leq j \leq k_0} \ln(w_j) + 1 \right) (\ln(w_{k_0+1}) + 1) = s \prod_{2 \leq j \leq k_0+1} \ln(w_j) + 1$$

This completes the induction step, and thus the lemma is proved. $\qquad \square$

In the following Proposition we show that the disagreement coefficient of a line tree that assigns $0$ to all samples is of $O(\ln^d w)$ among line trees.

*Proof of Proposition 3.5.* Assume a line tree $h \in \mathbb{L}_{d'}$ that assigns the same label to all data points. Without loss of generality, assume this label is $0$. We want to bound $\theta(h) = \sup_r \frac{|\mathrm{DIS}(B_{\mathbb{L}_{d'}}(h,r))|}{nr}$. Fixing $r$, from Lemma D.1, we have:

$$|\mathrm{DIS}(B_{\mathbb{L}_{d'}}(h,r))| = \{x \mid \exists h' \in B_{\mathbb{L}_{d'}}(h,r) : h'(x) \neq h(x)\} = \{x \mid \exists h' \in B_{\mathbb{L}_{d'}}(h,r) : h'(x) = 1\}$$

Defining $F$ as the set of all possible functions $f_{h'}$ (See Definition D.8), we have $|F| = 2^{|d'|}$. We break $\mathrm{DIS}(B_L(h,r))$ based on $f_{h'}$:

$$\mathrm{DIS}(B_L(h,r)) = \bigcup_{f \in F} \{x \mid \exists h' \in B_{\mathbb{L}_{d'}}(h,r) \wedge f_{h'} = f : h'(x) = 1\}$$

If we define

$$A_{f,l} := \{x \mid \exists h' \in B_{\mathbb{L}_{d'}}(h,r) \wedge f_{h'} = f \wedge l_{h'} = l : h'(x) = 1\}$$

Then we have

$$\mathrm{DIS}(B_L(h,r)) = \bigcup_{f \in F} A_{f,0} + A_{f,1}$$

Therefore,

$$|\mathrm{DIS}(B_L(h,r))| \leq \sum_{f \in F} |A_{f,0}| + |A_{f,1}| \qquad (9)$$

We bound size of $A_{f,0}$ and $A_{f,1}$ separately.

**Bounding the Size of $A_{f,0}$**

We aim to bound the cardinality of the following set:

$$A_{f,0} = \left\{ x \mid \exists h' \in B_{\mathbb{L}_{d'}}(h,r), \; f_{h'} = f, \; l_{h'} = 0 : \; h'(x) = 1 \right\}. \qquad (10)$$

Recall that $B_{\mathbb{L}_{d'}}(h,r)$ denotes the ball of radius $r$ around $h$ within the class of Line Trees of depth at most $d'$.

Because $h' \in B_{\mathbb{L}_{d'}}(h,r)$, it follows that $D(h,h') \leq r$, meaning that $h'$ differs from $h$ on at most an $r$ fraction of the $n$ total points. Since the original classifier $h$ assigns label $0$ to every point, $h'$ can label at most $nr$ points as $1$. Equivalently,

$$|\{x \mid h'(x) = 1\}| \leq nr \Rightarrow |\{x \mid h'(x) = 0\}| \geq n(1-r).$$

From Lemma D.10, the number of points classified as $0$ by $h'$ can be expressed as:

$$n(1-r) \leq |\{x \mid h'(x) = 0\}| = \prod_{a \in d'} h'^a \cdot \prod_{a \notin d'} w_a,$$

where $w_a$ is the width in dimension $a$.

Now, consider a point $x$ that is labeled $1$ by $h'$. By the structure of the tree, since $l_{h'} = 0$ there must exist a node corresponding to a dimension $b \in d'$ that routes $x$ contrary to the main path. Note

that $t_{h',b}$ denotes the threshold applied to dimension $b$. Samples with $x_b \leq t_{h',b}$ are directed to the left child, while those with $x_b > t_{h',b}$ are routed to the right child. So to bound the Size of $A_{f,0}$ we future break the set (Equation 10) based on dimension of the node that sample $x$ leave the path toward $l_h = 0$. Assuming this is dimension $b$, we consider two cases, depending $f_{h'}(b)$:

- $f_{h'}(b) = \text{prefix}$

  Here, $h'^b = t_{h',b} - 1$, and $h'(x) = 1$ only if $t_{h',b} \leq x_b$. Therefore,

  $$\prod_{a \notin d'} w_a \prod_{a \in d'} h'^a \leq (x_b - 1) \prod_{a \notin d'} w_a \prod_{a \in d' \setminus \{b\}} h'^a.$$

  Since for all $a \in d' \setminus \{b\}$, $h'^a \leq w_a$, and $\prod_{a=1}^{\dim} w_a = n$, we have $\prod_{a \notin d'} w_a \cdot \prod_{a \in d' \setminus \{b\}} w_a = \frac{n}{w_b}$.. Putting it all together, we have

  $$n(1 - r) \leq (x_b - 1)\frac{n}{w_b}$$

  which rearranges to

  $$w_b(1 - r) + 1 \leq x_b.$$

  Thus, for fixed $b$, the number of possible values for $x_b$ is at most

  $$w_b - (w_b(1 - r) + 1) + 1 = w_b r,$$

  and, for each fixed $b$, the number of possible $x$ is

  $$w_b r \prod_{a \in d'/\{b\}} w_a \prod_{a \notin d'} w_a = nr.$$

- $f_{h'}(b) = \text{suffix}$

  Now, $h'^b = w_b - t_{h',b} + 1$, and for $h'(x) = 1$, we require $x_b < t_{h',b}$, so

  $$h'^b \leq w_b - x_b.$$

  Therefore,

  $$\prod_{a \notin d'} w_a \prod_{a \in d'} h'^a \leq (w_b - x_b) \prod_{a \notin d'} w_a \prod_{a \in d' \setminus \{b\}} w_a.$$

  By using the same bounding and product arguments as above,

  $$n(1 - r) \leq (w_b - x_b)\frac{n}{w_b}$$

  which simplifies to

  $$x_b \leq w_b r.$$

  Thus, there are at most $w_b r$ such $x_b$, yielding at most $nr$ points in total for a fixed $b$.

Across all possible choices of $b \in d'$, the total number of such points $x$ is at most

$$\sum_{b \in d'} nr = nr|d'| \leq nr \cdot d$$

where $d$ is the depth of the Line Tree, i.e., $|d'| \leq d$.

*Thus,*

$$|A_{f,0}| \leq dnr.$$

**Bounding the Size of $A_{f,1}$**

We aim to bound the cardinality of the following set:

$$A_{f,1} = \left\{ x \mid \exists h' \in B_{\mathbb{L}_{d'}}(h, r),\ f_{h'} = f,\ l_{h'} = 1 :\ h'(x) = 1 \right\}.$$

According to Lemma D.11, if $x$ is classified as 1 by a line tree $h'$ with parameters $f_{h'}$ and $l_{h'} = 1$, then every point $x'$ satisfying the following will also be classified as 1 by $h'$:

$$\begin{cases} x'_a \leq x_a & \text{if } f_{h'}(a) = \text{prefix} \\ x_a \leq x'_a & \text{if } f_{h'}(a) = \text{suffix} \end{cases} \quad \forall a \in d_{h'},$$

we have $h(x') = l_h$. This describes a corner-aligned box in the input space whose size depends on $x$ and the direction assignments $f_{h'}$.

To express the size of the box leading to 1 under $h'$, define

$$x^a = \begin{cases} x_a & \text{if } f_{h'}(a) = \text{prefix}, \\ w_a - x_a + 1 & \text{if } f_{h'}(a) = \text{suffix}, \end{cases} \quad \forall a \in d_{h'}.$$

By this definition, for each $x$ that is classified as 1, the region of points labeled 1 under $h'$ contains at least $\prod_{a \notin d'} w_a \prod_{a \in d'} x^a$ distinct data points.

Since all such $h'$ under consideration are within distance $r$ from $h$, which labels all points as 0, the number of points for which $h'$ differs from $h$ (i.e. the number of points classified as 1 by $h'$) is at most $nr$. That is, $\sum_{x \in S} \mathbb{I}(h'(x) = 1) \leq nr$. Therefore, for any $x$ such that $h'(x) = 1$, the box it produce as above must satisfy

$$\prod_{a \notin d'} w_a \prod_{a \in d'} x^a \leq nr.$$

For a fixed assignment of $f_{h'}$, values for dimensions $a \in d'$ is uniquely determined by the tuple $\{x^a : a \in d'\}$; that is, knowing these values and the directions and $x_a$ for $a \notin d'$ fixes $x$.

We now seek to bound the number of tuples $(x^a)_{a \in d'}$ such that $\prod_{a \in d'} x^a \leq nr$, with each $x^a$ an integer in $[1, w_a]$.

By Lemma D.12, the number of integer tuples $(x^a)_{a \in d'}$ that satisfy $\prod_{a \notin d'} w_a \prod_{a \in d'} x^a \leq nr \Rightarrow \prod_{a \in d'} x^a \leq nr \frac{1}{\prod_{a \notin d'} w_a}$ is at most

$$nr \frac{1}{\prod_{a \notin d'} w_a} \prod_{a \in d'} \ln(w_a) + 1 \leq nr \frac{1}{\prod_{a \notin d'} w_a}(1 + \ln w)^d.$$

Since the number of ways we can fix $x_a$ for $a \notin d'$ is, $\prod_{a \notin d'} w_a$ we have:

$$|A_{f,1}| \leq nr (\ln w + 1)^d.$$

**Combining two above cases**

Combining two above cases and Equation 9, we have

$$|\text{DIS}(B_L(h, r))| \leq \sum_{f \in F} |A_{f,0}| + |A_{f,1}| \leq \sum_{f \in F} nrd + nr(\ln w + 1)^d \leq 2^d \cdot nr(d + (\ln w + 1)^d)$$

Therefore

$$\theta_h \leq 2^d(d + (\ln(w) + 1)^d) = (2\ln(w) + 2)^d + 2^d d$$

For $d \geq 2$ and $w \geq 8$ this is of

$$\theta_h \leq O\left((3 \ln w)^d\right)$$

$\square$

## D.1 PROOF OF COROLLARY 1.3

*Proof of Corollary 1.3.* Using the calculated disagreement coefficient of decision trees in Theorem 1.1 which is $\ln(n)^d$ and $V_H$ of decision tree in Lemma A.3 which is $2^d(d + \ln \dim)$, we can plug in these values to Theorem 1.2 which results in

$$\ln(n)\ln(n)^{2d}\left(2^d(d+\ln\dim)d\ln\ln n + \ln\frac{\ln n}{\delta}\right) + \frac{\ln(n)^{2d}}{\epsilon^2}\left(2^d(d+\ln\dim)\ln\frac{\ln(n)^d}{\epsilon} + \ln\frac{1}{\delta}\right)$$

This is of

$$O\left(\ln^{2d+2}(n)\left(2^d(d+\ln\dim)d + \ln\frac{1}{\delta}\right) + \frac{\ln^{2d}(n)}{\epsilon^2}\left(2^d(d+\dim)\ln\frac{\ln^d(n)}{\epsilon} + \ln\frac{1}{\delta}\right)\right)$$

$\square$

### D.2 Proof of Lower bound of $\Theta$

*Proof of Theorem 1.1 lowerbounds.* We establish the lower bound by analyzing the all-zero classifier, $h_0$. From the definition of the disagreement coefficient (definition 3.3), we have $\theta = \sup_{h\in H}\theta_h \geq \theta_{h_0}$, where $\theta_{h_0} = \sup_r \frac{|\mathrm{DIS}(B_H(h_0,r))|}{nr}$. We select the specific radius $r = \frac{w^{\dim-d+1}}{4n}$, which gives the bound:

$$\theta \geq \frac{|\mathrm{DIS}(B_H(h_0,r))|}{n\cdot r} = 4\frac{|\mathrm{DIS}(B_H(h_0,r))|}{w^{\dim-d+1}}$$

The $n = w^{\dim}$ data points exist on a dim-dimensional grid. We partition this grid into $2^{\dim}$ equal-sized orthants, assigning each point to its nearest "corner." By the symmetry of the grid, the number of points in the disagreement region, $|\mathrm{DIS}(B_H(h_0,r))|$, is identical for each orthant. We can therefore find the count for one orthant and multiply the result by $2^{\dim}$. We will analyze the orthant where all $x_i \leq w/2$.

To find a lower bound on this count, we focus on a specific, symmetric subset of points $x$ in this orthant that have $d$ dimensions with $x_i \leq w/4$ and $\dim - d$ dimensions with $w/4 < x_i \leq w/2$. By symmetry, there are $\binom{\dim}{d}$ such subsets. We can count the points in just one—where $x_1, \ldots, x_d \leq w/4$ and $x_{d+1}, \ldots, x_{\dim} \in (w/4, w/2]$—and multiply by this binomial coefficient.

A point $x$ is in $\mathrm{DIS}(B_H(h_0,r))$ if there exists a classifier $h' \in B_H(h_0,r)$ such that $h'(x) \neq h_0(x)$, meaning $h'(x) = 1$ (definition 3.2). For any such $x$, we can construct a "line tree" $h'$ of height $d$ that uses the first $d$ dimensions and classifies a point $y$ as 1 if and only if $y_i \leq x_i$ for all $i = 1, \ldots, d$. The distance $D(h_0, h')$ is the fraction of points $h'$ classifies as 1. The number of points classified as 1 is $(\prod_{i=1}^{d} x_i) \cdot w^{\dim-d}$, so the distance is:

$$D(h_0,h') = \frac{(\prod_{i=1}^{d} x_i)\cdot w^{\dim-d}}{n} = \frac{(\prod_{i=1}^{d} x_i)\cdot w^{\dim-d}}{w^{\dim}} = \frac{\prod_{i=1}^{d} x_i}{w^d}$$

We only count points $x$ for which $h'$ is in the ball, $D(h_0, h') \leq r$. Substituting our values for $D(h_0, h')$ and $r$:

$$\frac{\prod_{i=1}^{d} x_i}{w^d} \leq \frac{w^{\dim-d+1}}{4n} \implies \prod_{i=1}^{d} x_i \leq \frac{w^{\dim+1}}{4n} = \frac{w}{4}$$

We can now find the total count. The number of sequences $\langle x_1, \ldots, x_d\rangle$ satisfying $x_i \leq w/4$ and $\prod_{i=1}^{d} x_i \leq w/4$ has a lower bound proportional to $w/4 \cdot \ln^{d-1}(w/4)$ (lemma D.13) Let $C' = \frac{1}{d!}$ be this constant of proportionality. The number of sequences $\langle x_{d+1}, \ldots, x_{\dim}\rangle$ with $w/4 < x_i \leq w/2$ is exactly $(w/4)^{\dim-d}$.

The total number of points in the disagreement region, $|\mathrm{DIS}|$, is lower-bounded by the product of our terms:

$$|\mathrm{DIS}| \geq \Omega\left((\text{Num. Orthants})\cdot(\text{Num. Groups})\cdot(\text{Count for } d \text{ dims})\cdot(\text{Count for } \dim - d \text{ dims})\right)$$

$$\geq \Omega\left(2^{\dim}\cdot\binom{\dim}{d}\cdot\left[C'(w/4)\ln^{d-1}(w/4)\right]\cdot\left[(w/4)^{\dim-d}\right]\right)$$

$$\geq \Omega\left(2^{\dim}\cdot\binom{\dim}{d}\cdot C'\cdot(w/4)^{\dim-d+1}\cdot\ln^{d-1}(w/4)\right)$$

$$\geq \Omega\left(2^{-\dim+2d+2}\cdot\binom{\dim}{d}\cdot C'\cdot w^{\dim-d+1}\cdot\ln^{d-1}(w/4)\right)$$

Simplifying and using $n = w^{\dim}$ and $4^{\dim} = 2^{2\dim}$: Using the lower bound $\binom{\dim}{d} \geq (\frac{\dim}{d})^d$:

$$|\text{DIS}| \geq \Omega\left(2^{-\dim+2d+2} \cdot (\frac{\dim}{d})^{d-1} \cdot C' \cdot w^{\dim-d+1} \cdot \ln^{d-1}(w/4)\right)$$

Finally, we substitute this back into our inequality for $\theta$:

$$\theta \geq \Omega\left(4\frac{|\text{DIS}(B_H(h_0, r))|}{w^{\dim-d+1}}\right)$$

$$= \Omega\left(2^{-\dim+2d+4} \cdot (\frac{\dim}{d})^{d-1} \cdot \frac{1}{d!} \cdot \ln^{d-1}(w/4)\right)$$

$$= \Omega\left(\frac{2^{-\dim+2d+4}}{d^{d-1}d!} \ln^{d-1}((w/4)^{\dim})\right)$$

$$= \Omega\left(\frac{2^{-\dim+2d+4}}{d^{d-1}d!} \ln^{d-1}\left(n/4^{\dim}\right)\right) \geq \Omega\left(\frac{2^{-\dim}}{d^{d-1}d!} \ln^{d-1}\left(n/4^{\dim}\right)\right)$$

So for $C = \frac{2^{-\dim}}{d^{d-1}d!}$ and $C' = \dim \ln 4$ we have

$$\theta \geq C \cdot (\ln(n) - C')^{d-1}.$$

$\square$

**Lemma D.13.** *Let $g(s, d)$ be the number of positive integer sequences $\langle x_1, \ldots, x_d \rangle$ such that the product satisfies $\prod_{i=1}^{d} x_i \leq s$. The count $g(s, d)$ has a lower bound*

$$g(s, d) = \Omega\left(\frac{s}{d!} \cdot \ln^{d-1}(s)\right)$$

*Proof of Lemma D.13.* Let $\tau_d(n)$ denote the $d$-th divisor function, defined as the number of ways to express a positive integer $n$ as a product of $d$ positive integers. We observe that the quantity $g(s, d)$ corresponds exactly to the summatory function of $\tau_d(n)$:

$$g(s, d) = \sum_{1 \leq n \leq s} \tau_d(n).$$

The asymptotic behavior of this sum is the subject of the Piltz Divisor Problem. It is a classical result in analytic number theory (Titchmarsh & Heath-Brown (1986)) that for $s \to \infty$:

$$g(s, d) = \sum_{n \leq s} \tau_d(n) = \frac{s(\ln s)^{d-1}}{(d-1)!} + O\left(s(\ln s)^{d-2}\right).$$

The leading term implies that for sufficiently large $s$, the count grows as the specified bound. Specifically, the main term dominates the error term, implying:

$$g(s, d) \geq C \cdot \frac{s(\ln s)^{d-1}}{(d-1)!}$$

for some constant $C > 0$. Thus, $g(s, d) = \Omega\left(\frac{s}{(d-1)!} \ln^{d-1}(s)\right)$. $\square$

### D.3 NECESSITY OF ASSUMPTIONS PROOFS

In this section, we provide the proofs from Section 3.3. Specifically, we first prove that if nodes are allowed to work on the same dimension as one of their ancestors, the disagreement coefficient is of $\Omega(n^{1/\dim})$

*Proof of Theorem 3.6.* We aim to construct a depth-2 decision tree that assigns label 1 to a point $x \in \mathbb{R}^{\dim}$ if and only if $x_a = c$, for some fixed dimension $a$ and constant $c \in \mathbb{R}$. The structure of such a tree is shown in Figure 2a, where the root node checks whether $x_a \geq c$ and the second node checks whether $x_a < c + 1$. Because both comparisons involve only the $a$-th coordinate, the tree assigns label 1 exactly to the inputs $x$ satisfying $x_a = c$, and label 0 otherwise.

Let $X \subset \mathbb{R}^{\dim}$ be a dataset of size $n$, and let the reference classifier $h_0$ be the constant-zero function. Consider the hypothesis ball $B_H(h_0, r)$ of radius $r = 2 \cdot n^{-1/\dim}$, which includes all classifiers that differ from $h_0$ on at most $2 \cdot n^{1-1/\dim}$ datapoints.

Fix a dimension $a \in [\dim]$. For each value $c$ that appears in the $a$-th coordinate of the dataset, define the set (or "row") $R_c^a := \{x \in X \mid x_a = c\}$. If $|R_c^a| \leq 2 \cdot n^{1-1/\dim}$, then the decision tree shown in Figure 2a labels only the points in $R_c^a$ as 1, and all others as 0. Such a classifier differs from $h_0$ on at most $2 \cdot n^{1-1/\dim}$ points and therefore lies in $B_H(h_0, r)$. Consequently, all points in such a row $R_c^a$ lie within the disagreement region $\mathrm{DIS}(B_H(h_0, r))$. We call such rows **light rows**.

Rows $R_c^a$ for which $|R_c^a| > 2 \cdot n^{1-1/\dim}$ are called **heavy rows**. We now upper-bound the number of heavy rows per dimension. Since each heavy row contains more than $2 \cdot n^{1-1/\dim}$ points, their total number for a fixed dimension $a$ is at most:

$$\frac{n}{2 \cdot n^{1-1/\dim}} = \frac{1}{2} n^{1/\dim}.$$

A point $x \in X$ can be excluded from the disagreement region only if it lies in a heavy row for **every** dimension. Formally, we have:

$$x \notin \mathrm{DIS}(B_H(h_0, r)) \quad \Rightarrow \quad \forall a \in [\dim], \ \exists c \text{ such that } x \in R_c^a \text{ and } |R_c^a| > 2 \cdot n^{1-1/\dim}.$$

Since there are at most $\frac{1}{2} n^{1/\dim}$ heavy rows in each dimension, the number of points that lie in a heavy row for all dim dimensions is at most:

$$\left( \frac{1}{2} n^{1/\dim} \right)^{\dim} = \frac{n}{2^{\dim}}.$$

Thus, at least

$$n - \frac{n}{2^{\dim}} = \frac{2^{\dim} - 1}{2^{\dim}} n$$

points belong to the disagreement region:

$$|\mathrm{DIS}(B_H(h_0, r))| \geq \frac{2^{\dim} - 1}{2^{\dim}} n.$$

The disagreement coefficient at radius $r = 2 \cdot n^{-1/\dim}$ is therefore lower bounded by:

$$\theta \geq \theta_{h_0} \geq \frac{|\mathrm{DIS}(B_H(h_0, r))|}{rn} \geq \frac{\frac{2^{\dim} - 1}{2^{\dim}} n}{2n^{1-1/\dim}} = \frac{2^{\dim} - 1}{2^{\dim+1}} n^{1/\dim} = \Omega(n^{1/\dim}).$$

This completes the proof. $\qquad \square$

Next we show that if there is no assumption on input dataset then, the disagreement coefficient is of $\Omega(n)$.

*Proof of Theorem 3.7.* Consider the dataset $X = \{X_i = (i, i, \ldots, i) \in \mathbb{N}^{\dim} \mid i = 1, \ldots, n\}$, where all data points lie along the diagonal line $x_1 = x_2 = \cdots = x_{\dim}$. Let the reference classifier $h_0$ be the constant-zero classifier, i.e., $h_0(x) = 0$ for all $x \in X$.

Let $r = \frac{1}{n}$. The hypothesis ball $B_H(h_0, r)$ contains all decision tree classifiers $h'$ such that $h'$ differs from $h_0$ on at most one point. Since each point $X_i$ is distinct and isolated, we can construct a tree $h_i \in B_H(h_0, r)$ that outputs $h_i(X_i) = 1$ and $h_i(x) = 0$ for all $x \neq X_i$.

To isolate a specific point $X_c = (c, c, \ldots, c)$ in the dataset, it is sufficient to use a decision tree of depth two that queries only the first two coordinates. This construction is illustrated in Figure 2b.

The tree works as follows:

1. The root node tests whether $x_1 \geq c$.

2. If not, the label is $0$.

3. Otherwise, the second node checks whether $x_2 < c + 1$.

4. If this is true, the label is $1$; otherwise, the label is again $0$.

Because each $X_i$ lies along the diagonal (i.e., $x_1 = x_2 = \cdots = x_{\dim} = i$), this tree correctly assigns label $1$ only to the point $X_c$, and label $0$ to all other $X_i$. Thus, even under the structural constraint that no dimension repeats along a path, we can construct such an isolating tree using only two features.

Hence, for every point $X_i$, there exists $h_i \in B_H(h_0, r)$ such that $h_i(X_i) \neq h_0(X_i)$, which implies that every point $X_i$ lies in the disagreement region:

$$\mathrm{DIS}(B_H(h_0, r)) = \{x \in X \mid \exists h' \in B_H(h_0, r) \text{ s.t. } h'(x) \neq h_0(x)\} = X.$$

Therefore,

$$|\mathrm{DIS}(B_H(h_0, r))| = n, \quad \text{and} \quad r = \frac{1}{n},$$

so the disagreement coefficient is at least

$$\theta_{h_0} = \sup_{r > 0} \frac{|\mathrm{DIS}(B_H(h_0, r))|}{rn} \geq \frac{n}{(1/n) \cdot n} = n.$$

This proves that the disagreement coefficient $\theta = \sup_{h \in H} \theta_h$ satisfies $\theta = \Omega(n)$.

$\square$

## D.4 RELAXING UNIFORMITY ASSUMPTION

In this section, we provide the method by which we relax the assumption of uniformity among samples. As outlined in the main body of the paper, we assign to each sample an importance measure, denoted as $1 \leq W_i \leq \lambda$. In this context, we evaluate the classifier's error using the formula:

$$\mathrm{err}_X^W(h) = \frac{\sum_{i=1}^n \mathbb{I}(h(X_i) \neq Y_i) W_i}{\sum_{i=1}^n W_i}.$$

We further define the distance between two classifiers in this weighted context with the following expression:

$$D^W(h_1, h_2) = \frac{\sum_{i=1}^n W_i \mathbb{I}(h_1(X_i) \neq h_2(X_i))}{\sum_{i=1}^n W_i}.$$

. Similarly, the ball $r$ of classifiers within around a given classifier $h$ is defined as:

$$B_H^W(h, r) = \{h' \in H \mid D^W(h, h') \leq r\}.$$

Similarly, the disagreement coefficient $\theta_h^W$ of classifier $h$ is defined as:

$$\theta_h^W = \sup_{0 < r} \frac{\sum_{i \in \mathrm{DIS}(B^W(h,r))} W_i}{r \sum_{i=1}^n W_i},$$

The following theorem establishes a bound on the disagreement coefficient for the weighted case, showing that it is at most $\lambda^2$ times the disagreement coefficient for the unweighted case.

**Theorem D.14.** *In any classification task where $1 \leq W_i \leq \lambda$ for all $i$, the disagreement coefficient $\theta_h^W$ is at most $\lambda^2$ times the disagreement coefficient $\theta_h$ in the case where all samples have equal weight.*

The proof of Theorem D.14 leverages the relationship between the weighted and unweighted distances between classifiers. Specifically, we show that the weighted distance $D^W$ is bounded by $\lambda$ times the unweighted distance $D$, i.e.,

$$D^W(h_1, h_2) \leq \lambda D(h_1, h_2)$$

This relationship implies that the set of classifiers $B_H^W(h, r)$ that are within a distance $r$ of classifier $h$ in the weighted case is a subset of the corresponding set in the unweighted case, $B_H(h, r\lambda)$. Therefore,

$$B_H^W(h, r) \subseteq B_H(h, r\lambda)$$

which helps us prove the theorem.

*Proof of Theorem D.14.* We aim to prove that for any classifier $h$ and radius $r$:

$$\lambda^2 \frac{|\text{DIS}(B_H(h, r\lambda))|}{n(r\lambda)} \geq \frac{\sum_{i \in \text{DIS}(B_H^*(h,r))} W_i}{r \sum_i W_i}$$

First, consider any two classifiers $h_1$ and $h_2$. The weighted distance $D^*(h_1, h_2)$ is given by:

$$D^*(h_1, h_2) = \frac{\sum_i W_i \mathbb{I}(h_1(X_i) \neq h_2(X_i))}{\sum_i W_i}$$

Since $W_i \leq \lambda$, we have:

$$D^*(h_1, h_2) \leq \lambda \frac{\sum_i \mathbb{I}(h_1(X_i) \neq h_2(X_i))}{\sum_i W_i}$$

Given $1 \leq W_i$, this further simplifies to:

$$D^*(h_1, h_2) \leq \lambda \frac{\sum_i \mathbb{I}(h_1(X_i) \neq h_2(X_i))}{n} = \lambda D(h_1, h_2)$$

From this, we observe that if $D^*(h_1, h_2) \leq r$, then $D(h_1, h_2) \leq \lambda r$. Therefore:

$$B_H^*(h, r) \subseteq B_H(h, r\lambda)$$

Given this inclusion, we have:

$$\frac{\sum_{i \in \text{DIS}(B_H^*(h,r))} W_i}{r \sum_i W_i} \leq \frac{\sum_{i \in \text{DIS}(B_H(h,\lambda r))} W_i}{r \sum_i W_i}$$

Since $1 \leq W_i \leq \lambda$, it follows that:

$$\frac{\sum_{i \in \text{DIS}(B_H(h,\lambda r))} W_i}{r \sum_i W_i} \leq \lambda \frac{\sum_{i \in \text{DIS}(B_H(h,\lambda r))} 1}{rn} = \lambda \frac{|\text{DIS}(B_H(h, \lambda r))|}{rn} = \lambda^2 \frac{|\text{DIS}(B_H(h, \lambda r))|}{(r\lambda)n}$$

Therefore:

$$\theta_h^* = \sup_{0 < r} \frac{\sum_{i \in \text{DIS}(B_H^*(h,r))} W_i}{r \sum_i W_i} \leq \lambda^2 \sup_{0 < r} \frac{|\text{DIS}(B_H(h, \lambda r))|}{(r\lambda)n} = \lambda^2 \theta_h$$

This completes the proof.

$\square$

# E  Additive Algorithms Are Insufficient in Multiplicative Settings

In this section, we examine why additive algorithms are fundamentally inadequate for multiplicative error settings. We outline two high-level approaches that one might consider when adapting additive algorithms for multiplicative guarantees, and demonstrate the inherent limitations of both.

- **Estimating the Optimal Error Rate (Without Output Verification):** A natural idea is to first estimate the optimal classifier's error, say to within a constant factor (for example, a 2-approximation), and then use this estimate as a baseline for additive algorithms. This strategy implicitly assumes either prior knowledge of or access to a tight estimate of the minimal achievable error. However, even if the optimal classifier is known, estimating its error rate $\eta$ to within a multiplicative factor of 2 with confidence $1 - \delta$ requires at least $\frac{1}{\eta} \ln \frac{1}{\delta}$ samples. Since $\eta$ is unknown in practice—and, crucially, should not appear in the label complexity of the final algorithm—this approach cannot yield a label-efficient algorithm for general $\eta$.

- **Verifying the Error Rate By Iterative Refinement:** Alternatively, one can attempt to iteratively refine the estimate of the optimal error. For instance, starting with a guess of $\eta = 1/2$, run the additive algorithm with $\epsilon' = \epsilon \cdot 1/2$. If this fails to yield the desired error guarantee, halve the estimate ($\eta = 1/4, \epsilon' = \epsilon \cdot 1/4$), and continue. This approach necessitates verifying, at each step, whether the returned classifier meets the guarantee $\mathrm{err}(h) \leq \eta + \epsilon'$. Such verification also requires $O\left(\frac{\ln(1/\delta)}{\eta}\right)$ labeled examples per attempt. As before, the unknown and potentially small value of $\eta$ causes the total label complexity to depend inversely on $\eta$, which is unacceptable in settings where $\eta$ is not known.

Both approaches—either **without verification** (relying on an accurate guess of $\eta$), or **with verification** (iteratively guessing and checking)—face the same fundamental barrier: The number of labeled examples needed to estimate or verify small error rates scales inversely with the (unknown) true error $\eta$. Because any algorithm that hopes to achieve a multiplicative error guarantee must operate efficiently even when $\eta$ is small and unknown, this unavoidable dependence is fatal. As a result, additive algorithms and their naive adaptations cannot provide effective or label-efficient solutions in the multiplicative error regime.

# F  Empirical Behavior of Algorithm Constants

We conducted experiments to assess the effect of algorithmic constants $c_1, b_1, c_2$, and $b_2$ in practical scenarios. Our key findings indicate that these constants can be set to relatively small values without adversely affecting the performance or correctness guarantees of Algorithm 1. In particular, we observed that values as low as 3 for $c_1, c_2, b_1$ and $b_2$ are sufficient in practice, despite theoretical analysis suggesting much larger values.

Additional experimental results, code and full raw data are available via our anonymous Dropbox link: http://bit.ly/458BS1r. Notably, our empirical results suggest that $c_1$ and $b_1$ have a larger impact on algorithm success rates compared to $c_2$ and $b_2$.

**Experimental setup:**

- Sample size: $n = 10^7$
- The optimal classifier was selected uniformly at random
- Labels were assigned according to the optimal classifier with independent label noise of 0.1
- Algorithm 1 was executed with $\delta = \epsilon = 0.1$
- Each configuration of $(c_1, b_1, c_2, b_2)$ was tested in 50 independent trials
- A configuration was considered successful if the success rate exceeded $1 - \delta = 0.9$
- Each experiment was ran on a single CPU core. Each setup takes 1 min to complete.

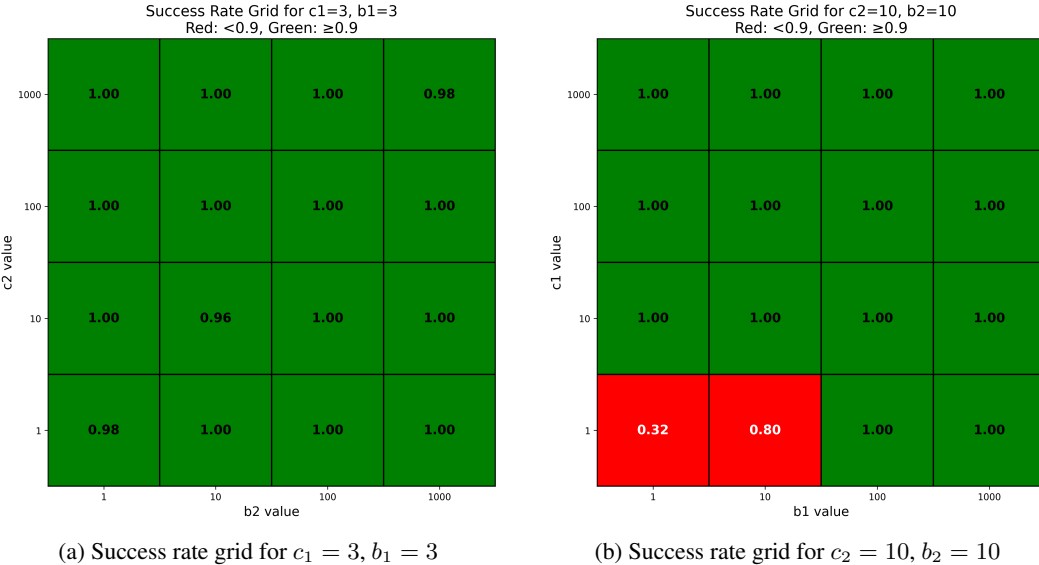

(a) Success rate grid for $c_1 = 3$, $b_1 = 3$        (b) Success rate grid for $c_2 = 10$, $b_2 = 10$

Figure 3: Comparison of success rate grids for various $(c_1, b_1, c_2, b_2)$ parameterizations when running Algorithm 1 with $\delta = 0.1$ (expected success rate $> 90\%$). For each cell we run Algorithm 1 on a fixed randomly generated dataset for 50 times and calculate the success rate. evidence suggests setting each of these constants to 3 is typically sufficient for reliable algorithm performance.

