# OpenReview forum: "Active Learning for Decision Trees with Provable Guarantees"
_ICLR.cc/2026/Conference — ICLR 2026 Poster_

### Official Review · Reviewer_RYtN · 2025-10-31

**Soundness:** 3
**Presentation:** 3
**Contribution:** 2
**Rating:** 6
**Confidence:** 4

**Summary:**

The paper considers the problem of active learning on decision trees. There are two main contributions of the paper. In the first contribution, the authors derive new bounds on the disagreement coefficient of the class of decision trees with a fixed depth and disjoint decision parameters at each node. This allows us to use existing algorithms and guarantees for active learning of decision trees that use the disagreement coefficient of the hypothesis class. In the second contribution, the authors propose a new algorithm for learning the optimal classifier upto a multiplicative error, as opposed to additive error, which is the common metric in learning theory. The authors provide proofs of their claims and combine the two contributions to obtain new algorithms with theoretical guarantees on sample complexity for active learning on decision trees.

**Strengths:**

I think the contribution of disagreement coefficient of decision trees is an interesting and useful contribution that can help us better understand learning using decision trees. Moreover, this will allow us to better contextualize decision trees among the other widely studied hypothesis classes for learning.

The multiplicative error bounds are also interesting, especially for the cases when the error of the true hypothesis is very small (or zero).

**Weaknesses:**

The paper does not have any glaring weakness but I do have some questions (see next section).

**Questions:**

I have several questions:

- I am still a bit confused about the setup. So in your setting, the learner has access to a fixed dataset $S$ consisting of $n$ unlabeled data points.  And whenever the learner wants the label of some point, it can query an oracle that returns a label $y \sim \mathbb{P}(y|x)$, based on some distribution?

- What does curly R in the algorithms denote? (This is likely somewhat related to the previous question)

- Theorem A.1. only holds when the distribution $\mathbb{P}(Y|X)$ is sub-Gaussian. In learning theory literature, this is more commonly referred to as the Massart noise condition. In this scenario $|\Pr(y|x) - 1/2| > \eta$ for some $\eta > 0$ and all $x, y$. This is something that should be clarified both in your upper and lower bound. There are also less benign noise conditions (Tsybakov conditions) where the dependence on $\varepsilon$ will get worse.

- The terms $\mathrm{err}(h)$ and $\widehat{\mathrm{err}}(h)$ have not been explicitly defined.

- How is the proposed algorithm for stump classifier different from a classical binary search (with repeated queries for noise)?

- In typical scenarios in active learning, it is assumed that the samples come from underlying distribution. In your setting, you have sort of fixed the distribution to the empirical dataset. While this might be fine, but does it not automatically restrict the number of hypotheses your hypothesis class? Does this not make your problem simpler in some sense?

---

> ### Author Response · Authors · 2025-11-20
>
> We thank the reviewer for their positive assessment, particularly for highlighting that our disagreement coefficient analysis allows for better contextualization of decision trees within learning theory, and for appreciating the multiplicative error bounds. We are pleased to clarify the setup and theoretical questions below.
>
> 1\. Clarifying the Noise Model (Theorem A.1)
>
> The noise conditions (like Massart or Tsybakov) are essential for deriving "fast rates" (where sample complexity scales with $O(1/\\epsilon)$). In this work, however, we utilize the standard VC-dimension uniform convergence bounds (e.g., Anthony & Bartlett, 2009\) which provide "slow rates" (scaling with $O(1/\\epsilon^2)$). These general bounds hold for any bounded loss function in the fully agnostic setting, without requiring specific assumptions on the noise distribution $P(Y|X)$. By targeting the $1/\\epsilon^2$ regime, we ensure our results remain valid without any assumption on the noise.
>
> 2\. Comparison with Binary Search (Question 5\)
>
> Standard binary search relies on the realizable assumption (a perfect threshold exists), where a single query tells you exactly which direction to move. In our agnostic setting, labels can be arbitrary. A single query cannot reliably determine the direction of the optimal threshold because the error surface is not convex or monotonic. Instead, our algorithm samples a batch of labels from the active region to estimate error rates. Based on these samples, it calculates confidence intervals (LB and UB) and prunes the interval $\[L\_i, R\_i\]$ from the left and right, removing thresholds that are statistically confident to be suboptimal (i.e., their lower bound error exceeds the minimum upper bound error). This statistical pruning allows us to converge even when the labels are from arbitrary distribution.
>
> 3\. The Setup: Fixed Dataset vs. Underlying Distribution (Question 1 & 6\)
>
> Our setting is pool-based active learning (specifically, the discrete setting used in works like Musco et al., 2022). We have a fixed pool of $n$ data points $S$. The "distribution" is the uniform distribution over $S$.
>
> While the domain is discrete, the problem is non-trivial. We are finding the hypothesis $h$ that minimizes the error on $S$. As we show in our new lower bound (Theorem 1.1), the complexity is dictated by the disagreement coefficient, which we prove can be as high as $\\Omega(\\ln(n)^d)$ even in this discrete setting.
>
> 4\. Notation Clarifications (Question 2 & 4\)
>
> We have updated the manuscript to explicitly define these terms to ensure clarity for the reader:
>
> * $\\mathcal{R}(a, S)$: We clarified in the text that this denotes taking a uniformly random subset of $S$ of size $a$ (sampling without replacement).
> * $err(h)$ vs $\\hat{err}(h)$: We added definitions stating that $err(h)$ is the true error on the current interval/dataset, while $\\hat{err}(h)$ is the empirical error calculated on the random subsample drawn during that specific iteration (used to construct the bounds LB and UB).
>
> We hope that our comments and revision address your concerns. If this is the case, we would be  grateful if you would potentially increase your score accordingly.
>
> **References**:
>
> 1. Anthony, Martin, and Peter L. Bartlett. *Neural network learning: Theoretical foundations*. Cambridge University press, 2009\.
>
> 2. Musco, Cameron, et al. "Active linear regression for $ℓ\_p$ norms and beyond." *2022 IEEE 63rd Annual Symposium on Foundations of Computer Science (FOCS)*. IEEE, 2022

---

### Official Review · Reviewer_gUF4 · 2025-10-31

**Soundness:** 3
**Presentation:** 3
**Contribution:** 2
**Rating:** 4
**Confidence:** 3

**Summary:**

This paper studies the classic pool-based active learning problem, where given a pool of unlabeled examples, one wants to design an adaptive algorithm that makes as few label queries over the pool as possible. This paper focuses on active learning for decision trees, a broadly studied class. Specifically, this paper contains two main results.
1. For structured decision trees and structured datasets, this paper develops the disagreement coefficients of the class of decision trees, using the framework developed by prior works, which gives a label complexity upper bound for active learning decision trees.
2. This paper generalizes the classic active learning framework, which considers additive error, to multiplicative error, and develops an algorithm for the upper bound.

These results are mathematically nontrivial, and the paper is written clearly.

**Strengths:**

Decision tree is an important and broadly studied class and the theory of active learning for decision trees has not been well understood yet. This paper makes progress in this direction by considering the disagreement coefficient framework proposed by Hanneke. For structured decision trees and structured distributions (uniform distribution over grid point), this paper gives an upper bound on the disagreement coefficient, which gives an upper bound on the label complexity of the problem. Furthermore, this paper also propose an algorithm that aims to achieve a multiplicative error instead of additive error, which is broadly considered in the literature. Overall, these results seem technically non-trivial and the proofs are complete.

**Weaknesses:**

Although technically solid, I am not sure if the contribution is significant enough.

1. For the disagreement coefficient of the decision trees, this paper places structural assumptions on both the type of decision trees and the datasets. This looks a bit too strong. In particular, if the marginal distribution is structured, then the disagreement coefficient does not characterize the min-max label complexity for the problem (for example, a halfspace has disagreement coefficient $\sqrt{d}$ under uniform distribution over the sphere, which leads to a label complexity $d^{3/2}$ using Hanneke's framework, but clearly there are algorithms with label complexity $O(d)$). This makes the motivation of bounding the disagreement coefficient a bit confusing, especially given that the bound is exponentially large and grows with respect to the size of the grid.

2. The motivation for considering multiplicative error is not very clear to me. Specifically, the label complexity upper bound for considering a multiplicative error depends on the size of the dataset $n$. In the realizable setting, a hypothesis with $(1+\epsilon)$ multiplicative error means it has $0$ prediction error over the distribution $D$, which statistically needs a dataset with unbounded size to learn, which makes the label complexity bound meaningless.

**Questions:**

1. This paper considers learning over a discrete dataset, which is a bit different from the classic generalization setting, where one has a dataset sampled from a distribution and one considers the generalization error over the distribution. This difference looks important, since the distribution considered in this paper is uniform over an integer grid and the disagreement coefficient depends on the size of the grid. What if we consider learning a decision tree over the uniform distribution over a box? Does this blow up the disagreement coefficient?

2. The above difference also makes learning upto a multiplicative error a bit confusing. Since a dataset is usually generated from a distribution, in the low-noise rate setting, it seems to me if we consider the generalization error, then learning to a $(1+\epsilon)opt$ error is not realistic, as it needs an unbounded number of samples to achieve this. Can you comment on this?

---

> ### Author Response · Authors · 2025-11-20
>
> We thank the reviewer for recognizing the technical soundness and clarity of the paper. We appreciate the challenging questions regarding the significance of our contributions and the choice of the pool-based setting. We believe that the following clarifications, particularly regarding our new lower bound results, address the concerns about the strength of our assumptions.
>
> 1\. Significance and Strength of Assumptions (Weakness 1\)
>
> * While the assumptions are indeed strong, Section 3.3 proves they are *necessary*. We show that relaxing them (e.g., allowing reused features or unstructured data) immediately leads to polynomial disagreement coefficients. Thus, these assumptions represent the "boundary of feasibility" for efficient active learning in this class.
> * Lower Bound (Theorem 1.1): To address this, we have added a rigorous lower bound in the revised paper. We prove that $\\theta \= \\Omega(c(\\ln(n)-c')^{d-1})$. This matches our upper bound up to a logarithmic factor.
> * Utility of $\\theta$: The disagreement coefficient remains the standard parameter characterizing the label complexity of active learning in the theoretical literature. Our work establishes the fundamental limits of this parameter for decision trees.
>
> 2\. Motivation for Multiplicative Error & Dependence on $n$ (Weakness 2 & Question 2\)
>
> * Pool-Based Setting: Our work operates in the fixed-pool (transductive) active learning setting, similar to recent theoretical works in active regression like Musco et al. (2022). In this setting, the goal is to label a *specific finite set of points* efficiently. In a finite pool of size $n$, achieving zero error (or multiplicative error) is possible with finite samples. Dependence on $\\ln(n)$ in our case.
> * We have three reasons for using multiplicative bounds.
>   1. Scale Invariance: It provides meaningful guarantees even when the optimal error is very small (where additive bounds would be loose).
>   2. Realizability: In the realizable case ($\\eta=0$), a multiplicative guarantee implies finding the perfect classifier, which additive bounds cannot guarantee without setting $\\epsilon \\approx 0$ (which blows up sample complexity).
>   3. Alignment with CS Theory: It bridges active learning with approximation algorithms and competitive analysis, where multiplicative factors are the standard.
>
> 3\. Continuous vs. Discrete Domains (Question 1\)
>
> Yes, if we switch to a continuous view the bounds become meaningless. The grid size $w$ essentially acts as a resolution parameter. As $w \\to \\infty$ (approaching the continuous case), our bounds scale with $\\ln(w)^d$. This implies that actively learning decision trees on continuous domains without further assumptions (like margin conditions) is unbounded in terms of $\\theta$. This aligns with the intuition that "arbitrarily fine" decision boundaries require infinite queries to resolve perfectly. Our work characterizes exactly *how* that complexity scales as the domain becomes more granular.
>
> We hope that our comments and revision address your concerns. If this is the case, we would be  grateful if you would potentially increase your score accordingly.

---

> > ### Author Response · Authors · 2025-11-27
> >
> > Dear Reviewer gUF4,
> >
> > As the discussion period draws to a close, we wanted to briefly follow up to see if you have any remaining questions or concerns regarding our rebuttal.
> >
> > We hope that our response has adequately addressed the issues raised in your review. If so, we would be grateful if you would consider updating your score accordingly.

---

> > ### Comment · Reviewer_gUF4 · 2025-11-27
> >
> > I would like to thank the authors for their responses. After reading the rebuttal, some of my concerns have been addressed, however, I am still a bit concerned about the implications of the work. For example, as mentioned in the rebuttal by the authors, when the domain becomes continuous, the bound provided by the paper becomes meaningless, which makes the results a bit restricted. I would like to raise my score, but I still feel the paper is around the borderline.

---

### Official Review · Reviewer_sttn · 2025-11-01

**Soundness:** 4
**Presentation:** 4
**Contribution:** 4
**Rating:** 8
**Confidence:** 3

**Summary:**

This paper studies the problem of actively learning decision trees and the problem of establishing multiplicative error bounds for general binary classification in active learning. The authors provide the first rigorous analysis of the *disagreement coefficient* for decision trees, a parameter that controls the label complexity in active learning. They bound this coefficient as $O(\ln^d(n))$, under some assumptions: (i) the input data must possess a regular, grid-like structure, and (ii) each root-to-leaf path in the decision tree must query distinct feature dimensions. The paper proves these assumptions are necessary; relaxing them leads to significantly higher (polynomial) label complexity. The authors introduce the first general active learning algorithm for binary classification that achieves a *multiplicative* error guarantee. This algorithm produces a $(1 + \epsilon)$-approximate classifier, meaning its error is at most $(1 + \epsilon)$ times the error of the optimal classifier. This framework offers stronger accuracy control than traditional additive error models. By combining these results, the authors design an active learning algorithm for decision trees that requires only a polylogarithmic number of label queries, provided the structural and data distribution assumptions hold.

**Strengths:**

- The results are interesting and make progress on important problems in learning theory.

- The techniques are interesting and non-trivial.

- Overall I found the paper well-written.

**Weaknesses:**

- The bounds are a bit unsatisfactory.

- The paper might be a bit hard to follow for some members of the ICLR community who are less on the theory side.

**Questions:**

- Is dependence on the VC dimension in Theorem 1.2 necessary?

- Is exponential dependence on $d$ in Corollary 1.3 necessary?

- The algorithm boxes are a bit notation heavy.

---

> ### Author Response · Authors · 2025-11-20
>
> We are sincerely grateful for the reviewer's assessment and for highlighting the non-trivial nature of our techniques. We are glad you found the paper well-written and the multiplicative error framework conceptually strong.
>
> We address your specific questions and comments below.
>
> 1\. On the "Unsatisfactory" Bounds & Exponential Dependence on $d$ (Question 2\)
>
> Yes, this dependence is unavoidable for decision trees.
>
> * **Theoretical Necessity:** The VC dimension of a decision tree of depth $d$ scales with $O(2^d)$. Since the sample complexity of any learning algorithm almost always scale linearly with the VC dimension (or the disagreement coefficient, which is related), the label complexity must inherently scale with $2^d$.
> * To rigorously confirm this, we have added Appendix D.2 in the revised paper. We prove a lower bound for the disagreement coefficient of $\\Omega(c(\\ln(n)-c')^{d-1})$ in theorem 1.1. This matches our upper bound up to a logarithmic factor, confirming that our derived rate is tight and the exponential dependence on $d$ is intrinsic to the hypothesis class, not an artifact of our analysis.
>
> 2\. Dependence on VC Dimension in Theorem 1.2 (Question 1\)
>
> Yes. In the standard PAC and active learning settings, sample complexity is generally lower-bounded by $\\Omega(V\_H)$ (or $\\Omega(V\_H/\\epsilon)$). An active learning algorithm can improve the dependence on $1/\\epsilon$ (e.g., to $\\ln(1/\\epsilon)$), but the dependence on the complexity of the hypothesis class ($V\_H$) typically remains linear.
>
> 3\. Notation Heavy Algorithm Boxes (Weakness & Question 3\)
>
> We appreciate this feedback. We agree that the algorithms require dense notation to remain mathematically precise (e.g., defining exact sampling sets and bounds). To improve readability without overcrowding the main text, we have:
>
> * Added a **Table of Notation** in the Appendix. This serves as a centralized lookup for symbols like $\\mathcal{R}$, $DIS(\\cdot)$, and $V\_H$, helping readers track definitions without flipping pages.
> * Explicitly clarified the definition of $\\mathcal{R}(a, S)$ (uniformly random subset sampling) in the text immediately preceding Algorithm 1 and 2, ensuring the specific symbol you questioned is defined locally.

---

### Official Review · Reviewer_5MpN · 2025-11-01

**Soundness:** 3
**Presentation:** 2
**Contribution:** 2
**Rating:** 4
**Confidence:** 4

**Summary:**

This paper investigates active learning for decision trees, providing theoretical bounds on the disagreement coefficient and proposing an analysis based on multiplicative error guarantees.

**Strengths:**

- The problem addressed is inherently interesting: active learning for non-linear, structured models like decision trees is a challenging and important direction.

- The choice to use multiplicative (rather than additive) bounds is conceptually appealing, as it aligns with realizable-case analyses and could, in principle, yield sharper guarantees.

**Weaknesses:**

- **Limited theoretical novelty and impact**
The theoretical contributions are incremental. The derived bounds on the disagreement coefficient $\theta = O((\ln n)^d)$ are relatively weak and pessimistic: $(\ln n)^d$ can grow very quickly even for moderately large depth, making the guarantee practically meaningless in realistic settings. Moreover, no *lower bounds* are provided, leaving it unclear whether the obtained rates are at all tight or informative.

- **Strong and unrealistic assumptions**
The analysis critically depends on highly restrictive assumptions: discrete and regular input domains, unique features per root-to-leaf path, and bounded tree depth. Such conditions are far from how decision trees are actually used in practice (e.g., CART, random forests, or gradient-boosted trees reuse features and operate over continuous domains). As a result, the results have very limited applicability.

- **Intractable use of the disagreement region**
The proposed Algorithm 2 relies on having access to disagreement regions. While this is standard in theoretical active learning, for decision trees the version space is exponentially large, and computing or even approximating $\text{DIS}$ is computationally hard. The paper does not propose any practical surrogate or heuristic for this step, making the algorithm purely theoretical and not implementable in realistic settings.

- **Lack of empirical validation**
While pure theoretical papers are completely acceptable, here the derived results are rather fragile (strong assumptions and exponential dependencies). On the other hand, no convincing experiments or simulations are provided to demonstrate the practical relevance of the theoretical claims. The “empirical study” mentioned in the appendix is neither summarized nor discussed in the main text. Without empirical evidence, it is hard to assess whether the proposed insights translate into meaningful performance gains.

- **Venue suitability**
This paper would be more suitable for a theoretical conference, where the focus on formal guarantees would be appreciated. However, even in that context, the work would likely be borderline due to the limited novelty, overly strong assumptions, and the lack of accompanying lower bounds.

While the topic is of potential interest, the present paper does not substantially advance the theory. The derived guarantees are fragile, the assumptions unrealistic, and the theoretical improvements marginal. Combined with the lack of lower bounds and the poor presentation quality, the work currently falls below the bar for a strong theoretical contribution.

**Questions:**

See the weaknesses section.

How do you define the calligraphic R appearing in Algorithm 1 and 2?

---

> ### Author Response · Authors · 2025-11-20
>
> We thank the reviewer for their time and for recognizing the inherent interest of active learning for decision trees and the conceptual appeal of our multiplicative error analysis. We appreciate the feedback regarding the lower bounds and empirical presentation.
>
> 1\. Theoretical Novelty & Lower Bounds (Addressing Weakness 1\)
>
> The reviewer expressed concern that the upper bounds are loose and that no lower bounds were provided. We have directly addressed this by adding a rigorous lower bound analysis in the revised paper (Theorem 1.1 and Appendix D.2).
>
> * We prove that the disagreement coefficient for decision trees is lower bounded by $\\Omega(c(\\ln(n)-c')^{d-1})$. Our upper bound is $O(\\ln^d n)$. This implies that our result is almost tight.
> * We emphasize that our analysis of the disagreement coefficient $\\theta$ (Section 3\) is a standalone contribution that applies to *any* disagreement-based active learning algorithm (additive or multiplicative), not just Algorithm 2\.
>
> 2\. Necessity of Assumptions (Addressing Weakness 2\)
>
> The reviewer noted that our assumptions (discrete grid, unique features) are restrictive. We agree these are strong assumptions; however, we prove they are theoretically necessary for efficient active learning, rather than artifacts of our analysis.
>
> * Impossibility Results: In Section 3.3, we formally prove that relaxing these assumptions destroys the polylogarithmic guarantees:
>   * Theorem 3.6: If features can be reused, $\\theta \= \\Omega(n^{1/dim})$.
>   * Theorem 3.7: Without the grid-like structure (i.e., on arbitrary input distributions), $\\theta \= \\Omega(n)$.
> * These theorems demonstrate a "phase transition" in learnability. We show that standard decision tree practices (reusing features, continuous domains) inherently lead to polynomial disagreement coefficient. Identifying these boundaries is a key contribution of this work.
>
> 3\. Intractability of Disagreement Regions (Addressing Weakness 3\)
>
> The reviewer correctly notes that computing the disagreement region is computationally hard.
>
> * Standard Theoretical Framework: We respectfully note that relying on the disagreement region is the standard framework for establishing statistical sample complexity bounds in active learning literature. Seminal works, such as Hanneke (2014), Balcan et al. (2006), and Dasgupta et al. (2007), utilize strictly analogous version space constructions to derive minimax rates.
> * Focus on Statistical Limits: Our primary goal is to determine the *information-theoretic* limits (sample complexity) of actively learning decision trees, which requires analyzing the version space. Heuristic approximations (like query-by-committee) are valuable for practice but do not provide the rigorous guarantees this paper seeks to establish.
>
> 4\. Empirical Validation (Addressing Weakness 4\)
>
> We apologize if the empirical section was previously easy to miss. We have revised Section 4.1 (Page 7\) to explicitly summarize our empirical findings and point to Appendix F for detailed results.  Our experiments demonstrate that Algorithm 1 performs reliably with constants significantly smaller than the worst-case theoretical bounds suggest (e.g., $c\_1, b\_1 \\approx 3$), supporting the practical feasibility of the method on simpler problems.
>
> 5\. Venue Suitability (Addressing Weakness 5\)
>
> We respectfully disagree that this work is unsuitable for ICLR. ICLR has a strong track record of publishing foundational learning theory, including specific advances in decision tree theory and active learning bounds. For example, “Average Sensitivity of Decision Tree Learning (ICLR 2023)”, “Sensitivity Verification for Additive Decision Tree Ensembles (ICLR 2025)” and “Improved Active Learning via Dependent Leverage Score Sampling (ICLR 2024)” are recent examples of ICLR papers focusing on rigorous theoretical guarantees for similar problems. Our work complements this tradition by establishing the first multiplicative error bounds for this class. Furthermore, decision trees have many practical applications in regression and classification, such as XGBoost and its variants like LightGBM, and their theoretical understanding is crucial to grasping fundamental problems and potentially improving them. As a result, many theoretical papers like this are published in main machine learning conferences like ICLR, ICML and NeurIPS. We omitted many additional examples due to space limits.
>
> 6\. Clarification on Notation
>
> Regarding the question on $\\mathcal{R}$ in Algorithms 1 and 2: We have clarified the definition in the text preceding the algorithms. $\\mathcal{R}(a, S)$ denotes a uniformly random subset of set $S$ of size $a$; that is, we sample $a$ distinct elements from $S$ without replacement.
>
> We hope that our comments and revision address your concerns. If this is the case, we would be  grateful if you would potentially increase your score accordingly.

---

> > ### Author Response · Authors · 2025-11-20
> >
> > **References:**
> >
> > 1. Dasgupta, Sanjoy, Daniel J. Hsu, and Claire Monteleoni. "A general agnostic active learning algorithm." *Advances in neural information processing systems* 20 (2007).
> > 2. Hanneke, Steve. "Theory of disagreement-based active learning." *Foundations and Trends in Machine Learning* 7.2-3 (2014): 131-309.
> > 3. Balcan, Maria-Florina, Alina Beygelzimer, and John Langford. "Agnostic active learning." *Proceedings of the 23rd international conference on Machine learning*. 2006
> > 4. Hara, Satoshi, and Yuichi Yoshida. "Average sensitivity of decision tree learning." *The Eleventh International Conference on Learning Representations*. 2023\.
> > 5. Ahmad, Arhaan, et al. "Sensitivity Verification for Additive Decision Tree Ensembles." *The Thirteenth International Conference on Learning Representations*. 2025\.
> > 6. Shimizu, Atsushi, et al. "Improved Active Learning via Dependent Leverage Score Sampling." *The Twelfth International Conference on Learning Representations*.

---

> > ### Comment · Reviewer_5MpN · 2025-11-25
> >
> > I thank the reviewers for the detailed answer.
> >
> > However, my concerns still remains.
> >
> > - The lower bound is of the form $(\log n - \text{dim})^d$ which is not very informative even in moderate dimensions.
> >
> > - As far as I know, finiteness (or any type of niceness) of the disagreement coefficient is not necessary for polylog savings in active learning. In that respect, the negative results provided in this work do not imply much in terms of learnability.

---

> > > ### Author Response · Authors · 2025-11-27
> > >
> > > We thank the reviewer for the continued engagement. We would like to address the two remaining points:
> > >
> > > 1. Lower Bound InformativenessRegarding the lower bound term $(\ln n - \text{dim})^d$, recall that the domain size is $n = w^{\text{dim}}$ (where $w$ is the grid width). Consequently, $\ln n = \text{dim} \cdot \ln w$. This means $\ln n$ is larger than $\text{dim}$ by a factor of $\ln w$. Therefore, for reasonably large $w$, this term is significant, making the lower bound close to our upper bound of $O(\ln^d n)$.
> > > 2. Relevance of Negative ResultsWhile we agree that the disagreement coefficient is not the sole determinant of learnability, it remains the central complexity measure for the canonical family of active learning algorithms, such as CAL and $A^2$. We are the first to strictly bound this parameter for decision trees, a contribution recognized as non-trivial by other reviewers. Our negative results formally delineate the limits of standard disagreement-based active learning for decision trees. This informs the community that to achieve polylogarithmic savings using these standard paradigms, specific structural assumptions (like those we analyze) are theoretically required, while also explicitly characterizing what is achievable under those assumptions.
> > >
> > > We hope this clarifies our contributions.

---

> ### Comment · Reviewer_5MpN · 2025-11-27
>
> 1. In many natural settings, the ambient dimension is large (i.e., the number of features is high), while each feature takes values in a fixed range. In that sense, I still find the lower bound somewhat loose for the regimes that are most relevant in practice.
>
> 2. I have not claimed that your result is trivial. My main concern is about its implications: as stated, it primarily rules out the applicability of disagreement-based algorithms, which in any case are not commonly used in practice. Indeed, several recent active learning methods do not rely on disagreement-based ideas (see the references below for some examples). A substantially stronger and more impactful result would be a lower bound that applies to any active learning algorithm.
>
> References
>
> Ashtiani, Kushagra, Ben-David. *Clustering with same-cluster queries*.
>
> Bressan, Cesa-Bianchi, Lattanzi, Paudice. *Margin-Based Active Learning of Classifiers*.
>
> Bressan, Cesa-Bianchi, Lattanzi, Paudice, J. Thiessen. *Active learning of classifiers with label and seed queries*.

---

> > ### Author Response · Authors · 2025-12-03
> >
> > Regarding the technical references provided, we must clarify that **none** of the papers you listed operate in the standard active learning setting where the learner queries a single data point and receives its label. This scenario remains the most natural setting, and most algorithms proposed for this setting are indeed disagreement-based. Our work addresses the fundamental limits of this standard setting, whereas the cited works achieve their results by altering the problem definition through stronger oracles or restrictive geometric assumptions:
> >
> > * **Geometric Assumptions:**
> >   * Ashtiani et al. (2016) rely on the strong $\\gamma$-margin property in Euclidean space, explicitly proving that without this specific geometric structure, the problem remains NP-hard.
> >   * Bressan et al. (2024) depend on "Convex Hull Margins" in $\\mathbb{R}^m$, a geometric separation assumption that does not hold for the general discrete or categorical domains.
> > * **Stronger Oracles:**
> >   * Bressan et al. (2022) utilize search oracles, which fundamentally simplify the problem by assuming the oracle can automatically locate positive examples within a subset, strictly exceeding the power of standard label verification.
> >
> > Therefore, our bounds remain significant for establishing the information-theoretic limits of the standard disagreement-based paradigm, which contains most algorithms proposed for standard active learning.
> >
> > Regrading your comment that disagreement-based algorithms are not used in practice, we would like to take this opportunity to respectfully reiterate that ICLR has a long history of publishing theoretical works, and it is not restricted to practical papers as suggested in your initial review and here.

---

### Author Response · Authors · 2025-12-03

We thank the reviewers for their insightful comments. The reviews recognize our work as the first rigorous analysis of the disagreement coefficient for decision trees and highlight the Theoretical significance of our multiplicative error framework.

**Contribution:  General Framework for Multiplicative Error** We wish to highlight a key contribution that extends beyond decision trees: we propose a general active learning algorithm that achieves multiplicative error guarantees for any classification task. Reviewers sttn, gUF4, and RYtN explicitly praised this contribution, calling it "conceptually appealing" and "mathematically non-trivial." Reviewer sttn noted that this is the "first general active learning algorithm" of its kind, marking a significant advance over the classic additive error framework.

**Major Improvement: Tight Lower Bounds** The main hesitation from reviewers (specifically 5MpN) was the lack of lower bounds to assess the tightness of our decision tree results. During the rebuttal, we derived and added a formal lower bound (Theorem 1.1) which matches our upper bound up to log factors. This confirms that the exponential dependence on dimension is intrinsic to the hypothesis class, not an artifact of our analysis.

**Addressing Theoretical Scope** Reviewers questioned the strength of our assumptions (discrete domains/unique features). We emphasized our negative results (Theorems 3.6 & 3.7), which prove that without these assumptions, active learning in this class requires polynomial samples. Our work therefore delineates the fundamental information-theoretic limits of active learning for decision trees.

**Conclusion:** With the strong support of Reviewer sttn (Score 8) and RYtN (Score 6), the positive shift in sentiment from gUF4 (who acknowledged concerns were addressed and raised their score during the discussion phase prior to ICLR’s reversion), and the addition of the lower bound theorem, we believe this paper provides a complete and rigorous foundation for the theory of active learning on decision trees, alongside a novel general framework for multiplicative error rate active learning.

---

### Meta-Review · Area_Chair_7THn · 2025-12-18

**Summary:**

The paper studies pool-based active learning of decision trees. The main result is a new label complexity upper bound relying on disagreement coefficient when certain assumptions on the concept class and data are made. During rebuttal, a lower bound was added. Many reviewers raised the concern that the upper bound is not satisfactory and that the assumptions are strong. The AC believes that this concern can be addressed in light of the lower bound.

**Reviewer Concerns:**

Reviewers' main concern is addressed.

**Reviewer Scores:**

Likely increase.

---

### Decision · Program_Chairs · 2026-01-26

Accept (Poster)